# DECONGESTION BY REPRESENTATION: LEARNING TO IMPROVE ECONOMIC WELFARE IN MARKETPLACES

**Omer Nahum**
Technion DDS

**Gali Noti**
Cornell CS

**David C. Parkes**
Harvard SEAS

**Nir Rosenfeld**
Technion CS

## ABSTRACT

Congestion is a common failure mode of markets, where consumers compete inefficiently on the same subset of goods (e.g., chasing the same small set of properties on a vacation rental platform). The typical economic story is that prices decongest by balancing supply and demand. But in modern online marketplaces, prices are typically set in a decentralized way by sellers, and the information about items is inevitably partial. The power of a platform is limited to controlling *representations*—the subset of information about items presented by default to users. This motivates the present study of *decongestion by representation*, where a platform seeks to learn representations that reduce congestion and thus improve social welfare. The technical challenge is twofold: relying only on revealed preferences from the choices of consumers, rather than true preferences; and the combinatorial problem associated with representations that determine the features to reveal in the default view. We tackle both challenges by proposing a *differentiable proxy of welfare* that can be trained end-to-end on consumer choice data. We develop sufficient conditions for when decongestion promotes welfare, and present the results of extensive experiments on both synthetic and real data that demonstrate the utility of our approach.

## 1 INTRODUCTION

Online marketplaces have become ubiquitous as our primary means for consuming tangible goods as well as services across many different domains. Examples span a variety of commercial segments, including dining (e.g., Yelp), real estate (e.g., Zillow), vacation homestays (e.g., Airbnb), used or vintage items (e.g., eBay), handmade crafts (e.g., Etsy), and specialized freelance labor (e.g., Upwork). A key reason underlying the success of these platforms is their ability to manage an exceptionally large and diverse collections of items, to which users are given immediate and seamless access. Once a desired item has been found on a platform then obtaining it should—in principle—be only 'one click away.'

But just like conventional markets, online markets are also prone to certain forms of market failure, which may hinder the ability of users to easily obtain valued items. One prominent type of failure, which our paper targets, is *congestion*. Congestion occurs when demand for certain items exceeds supply; i.e., when multiple users are interested in a single item of which there are not sufficiently-many copies available. E.g., in vacation rentals, the same vacation home may draw the interest of many users, but only one of them can rent it. This can prevent potential transactions from materializing, resulting in reduced social welfare—to the detriment of users, suppliers, and the platform itself.

In conventional markets, the usual economic response to congestion is to set prices in an appropriate manner (e.g., [26]). In our example, if the attractive vacation home is priced correctly, then only one user (who values it most, relative to other properties) will choose it; similarly, if other items are also priced correctly in relation to user valuations, then prices can fully decongest the market and the market can obtain optimal welfare, defined as the sum of users' valuations to their assigned items.

But for modern online markets, this approach is unattainable for two reasons. The first reason is that many online platforms do not have control over prices, which are instead set in a decentralized way by different sellers. The second reason is more subtle, but central to the solution we advance in this paper: we argue that an inherent aspect of online markets is that users make choices under limited information, and that this limits the effectiveness of price. Online environments impose natural constraints on the amount of information consumed, due to technical limitations (e.g., restricted screen space), behavioral mechanisms (e.g., cognitive capacity, attention span, impatience), or design choices (e.g.,

what information is highlighted, appears first as a default, or requires less effort to access). As such, the decisions of users are more affected by whatever information is made more readily available to them. From an economic perspective, this means they are making decisions under 'incorrect' preference models, for which (i) prices that decongest (or clear) at these erroneous preferences are incorrect, and (ii) prices that decongest at correct preferences still leave congestion at erroneous preferences.[1]

Partial information is therefore a reality that platforms must cope with—a new reality which requires new approaches. To ease congestion and improve welfare, our main thesis is that platforms can—and *should*—utilize their control over *information*, and in particular, on how items are represented to users. The decision of *representation*—the default way in which items are shown to users—is typically in the hands of the platform; and while providing equal access to all information may be the ideal, reality dictates that choosing *some* representation is inevitable.[2] Given this, we propose to use machine learning to solve the necessary design problem of choosing beneficial item representations.

To this end, we present a new framework for learning item representations that reduce congestion and promote welfare. Since congestion results from users making choices independently according to their own individual preferences, to decongest, the platform must act to (indirectly) coordinate these idiosyncratic choices; and since representations affect choices by shaping how users 'perceive' value, we will seek to coordinate perceived preferences. The basic premise of our approach is that, with enough variation in true user preferences, it should be possible to find representations for which choices made under perceived values remain both valuable *and* diverse. For example, consider a rental unit represented as having 'sea view' and 'sunny balcony' and draws the attention of many users but does not convey other information such as 'noisy location'; if users vary enough in how they value quietness, then showing 'quiet' instead of 'balcony' may help reduce congestion and improve outcomes.

From a learning perspective, the fundamental challenge is that welfare itself (and its underlying choices) depends on private user preferences. For this, we develop a proxy objective that relies on observable choice data alone, and optimizes for representations that encourage favorable decongested solutions through users' choices. A technical challenge is that representations are combinatorial objects, corresponding to a subset of features to show. Building on recent advances in differentiable discrete optimization, we modify our objective to be differentiable, thus permitting end-to-end training using gradient methods. To provide formal grounding for our approach of decongestion by representation, we theoretically study the connection between decongestion and welfare. Using competitive equilibrium analysis, we give several simple and interpretable sufficient conditions under which reducing congestion provably improves welfare. Intuitively, this happens when it is possible to present item features across which user preferences are more diverse, while at the same time hiding features that are not too meaningful for the users. The conditions provide basic insight as to when our approach works well.

We end with an extensive set of experiments that shed light on our proposed setting and learning approach. We first make use of synthetic data to explore the usefulness of decongestion as a proxy for welfare, considering the importance of preference variation, the role of prices, and the degree of information partiality. We then use real data of user ratings to elicit user preferences across a set of diverse items. Coupling this with simulated user behavior, we demonstrate the susceptibility of naïve prediction-based methods to harmful congestion, and the ability of our congestion-aware representation learning framework to improve economic outcomes. Code for all experiments can be found at: https://github.com/omer6nahum/Decongestion-by-Representation.

## 1.1 RELATED WORK

There is a growing recognition that many online platforms provide economic marketplaces, and considerable efforts have been dedicated to studying the role of recommender systems in this regard [7,25,27]. Some work, for example, has studied the effects of learning and recommendation on the equilibrium dynamics of two-sided markets of users and suppliers [5,21,13], exchange markets [10], or markets of competing platforms [6,14]. The main distinction is that our paper studies not what to show to users, but how. One study examined the effect of the complete absence of knowledge about some items on welfare [8]; in contrast, we study how welfare is affected by partial information about items. There are also studies on the role of information in the form of recommendations in enhancing

---

[1]Economic theory has many examples of other ways in which partial information hurts markets (e.g., [1]).
[2]In the influential book 'Nudge', Thaler and Sunstein (2008) argue similarly for 'choice architecture' at large.

system-wide performance with learning users; e.g., the use of selective information reporting to promote exploration by a group of users [18,20,3]. Again, this is quite distinct from our setting.

Conceptually related to our work is research in the field of human-centered AI [24] that studies AI-aided human decision making, and in particular prior work that has considered methods to learn representations of inputs to decision problems to aid in single-user decision making [12]. Related, there is work on selectively providing information in the form of advice to a user in order to optimize their decision performance [23]. It has also been argued that providing less accurate predictive information to users can sometimes improve performance [4]. These works, however, do not consider interactions between multiple users which are at the center of the types of markets we consider here.

Though underexplored in online markets, several works in related fields have considered how representations affect decisions. For example, [17] aim to establish the role of 'simplicity' in decision-making aids, and in relation to fairness and equity. Works in strategic learning have emphasized the role of users in representations; i.e., in learning to choose in face of strategic representations [22], and as controlling representations themselves [19]. Here we extend the discussion on representations to markets.

## 2 PROBLEM SETUP

The main element of our setting is a *market*, where each market is composed of $m$ indivisible items and $n$ users. Within a market, items $j$ are described by non-negative feature vectors $x_j \in \mathbb{R}_+^d$ and prices $p_j \geq 0$. Let $X \in \mathbb{R}^{m \times d}$ denote all item features, and $p \in \mathbb{R}^m$ denote all prices, which we assume to be fixed.[3] We mostly consider unit supply, in which there is only one unit of each item (e.g., as in vacation homes), but note our method directly extends to general finite supply, which we discuss later.

Each user $i$ in a market has a *valuation function*, $v_i(x)$, which determines their true value for an item with feature vector $x$. We use $v_{ij}$ to denote user $i$'s value for the $j$th item. We model each user with a non-negative, linear preference, with $v_i(x) = \beta_i^\top x$ for some *user type*, $\beta_i \geq 0$. The effect is that $v_i(x) \geq 0$ for all items, and all item attributes contribute positively to value. We assume w.l.o.g. that values are scaled such that $v_i(x) \leq 1$. Users have unit demand, i.e., are interested in purchasing a single item. Given full information on items, a rational agent would choose $y_i^* = \operatorname{argmax}_j v_{ij} - p_j$.

**Partial information.** The unique aspect of our setup is that users make choices on the basis of partial information, over which the system has control. For this, we model users as making decisions on an item with feature vector $x$ based on its *representation* $z$, which is truthful but lossy: $z$ must contain only information from $x$, but does not contain all of the information. We consider this to be a necessary aspect of a practical market, where users are unable to appreciate all of the complexity of goods in the market. Concretely, $z$ reveals a subset of $k \leq d$ features from $x$, where the set of features is determined by a binary *feature mask*, $\mu \in \{0,1\}^d$, with $|\mu|_1 = k$, that is fixed for all items. Each mask induces *perceived values*, $\tilde{v}$, which are the values a user infers from observable features:

$$\tilde{v}_i(x) = \beta_i^\top (x \odot \mu) = (\beta_i)_\mu^\top z, \tag{1}$$

where $\odot$ denotes element-wise product, and $(\beta)_\mu$ is $\beta$ restricted to features in $\mu$. For market items $x_j$ we use $\tilde{v}_{ij} = \beta_i^\top (x_j \odot \mu)$. Given this, under *partial* information, user $i$ makes choices $y_i$ via:[4]

$$y_i(\mu) = \texttt{choice}(X, p; v_i \mu) := \operatorname{argmax}_j \tilde{v}_{ij} - p_j, \tag{2}$$

where $\tilde{v}_{ij} - p_j$ is agent $i$'s perceived utility from item $j$, with $y_i(\mu) = 0$ encoding the 'no choice' option, which occurs when no item has positive perceived utility. Note Eq. (2) is a (simple) special case of [30]. When clear from context, we will drop the notational dependence on $\mu$. We use $y \in \{0,1\}^{n \times m}$ to describe all choices in the market, where $y_{ij} = 1$ if user $i$ chose item $j$, and 0 otherwise.

Under Eq. (2), each user is modeled as a conservative boundedly-rational decision-maker, whose perception of value derives from how items are represented, and in particular, by which features are shown. Note that together with our positivity assumptions, this ensures that representations cannot be used to portray items as more valuable than they truly are—which could lead to choices with negative utility.

---

[3]For example, this is reasonable when sellers adapt slowly (or not at all; e.g., as in ad auctions [9,2]), or when prices are set for a broader aggregate market. For discussion on adaptive prices, see Appendix A.

[4]In Appx. F.1 we experiment with an alternative decision model, which shows qualitatively similar results.

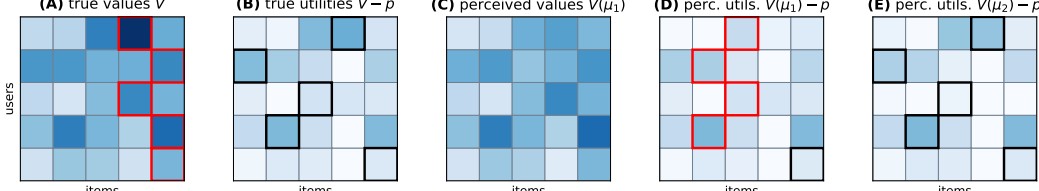

Figure 1: Values, prices, and choices. **(A)** A matrix $V$ of user-item values $v_{ij}$ in a market. User choices naturally congest (red squares), but at full information can be decongested with prices (**(B)**; black squares). Partial information may distort values only mildly (**(C)**, vs. (A)), but still deem prices as ineffective **(D)**. Nonetheless, some representations are better than others—and those we seek **(E)**.

**Allocation.** To model the effect of congestion we require an *allocation mechanism*, denoted $a = \texttt{alloc}(y_1, \dots, y_n)$, where $a \in \{0,1\}^{n \times m}$ has $a_{ij} = 1$ if item $j$ is allocated to user $i$, and 0 otherwise. We will use $a(\mu)$ to denote allocations that result from choices $y(\mu)$. We require *feasible allocations*, such that each item is allocated at most once and each user receives at most one item. For the allocation mechanism, we use the *random single round* rule, where each item $j$ in demand is allocated uniformly at random to one of the users for which $y_i = j$. This models congestion: if several users choose the same item $j$, then only one of them receives it while all others receive nothing. Intuitively, for welfare to be high, we would like that: (i) allocated items give high value to their users, and (ii) many items are allocated. As we will see, this observation forms the basis of our approach.

**Learning representations.** To facilitate learning, we will assume there is some (unknown) distribution over markets, $\mathcal{D}$, from which we observe samples. In particular, we model markets with a fixed set of items, and users sampled iid from some pool of users. For motivation, consider vacation rentals, where the same set of properties are available each week, but the prospective vacationers differ from week to week. Because preferences $\beta_i$ are private to a user, we instead assume access to *user features*, $u_i \in \mathbb{R}^{d'}$, which are informative of $\beta_i$ in some way. Letting $U \in \mathbb{R}^{n \times d'}$ denote the set of all user features, each market is thus defined by a tuple $M = (U, X, p)$ of users, items, and prices.

We assume access to a sample set $\mathcal{S} = \{(M^{(\ell)}, y^{(\ell)})\}_{\ell=1}^{L}$ of markets $M_\ell = (U^{(\ell)}, X, p^{(\ell)}) \sim \mathcal{D}$ and corresponding user choices $y^{(\ell)}$. Note this permits item prices to vary across samples, i.e., $p^{(\ell)}$ can be specific to the set of users $U^{(\ell)}$. Our overall goal will be to use $\mathcal{S}$ to learn representations that entail useful decongested allocations, as illustrated in Figure 1. Concretely, we aim for optimizing the *expected welfare* induced by allocations, i.e., the expected sum of values of allocated items:

$$W_{\mathcal{D}}(\mu) = \mathbb{E}_{\mathcal{D}}\Big[\sum_{ij} a(\mu)_{ij} v_{ij}\Big], \quad a(\mu) = \texttt{alloc}(y_1, \dots, y_n), \quad y_i = \texttt{choice}(X, p; v_i, \mu) \quad (3)$$

where expectation is taken also w.r.t. to possible randomization in `alloc` and `choice`. Thus, we wish to solve $\text{argmax}_\mu W_{\mathcal{D}}(\mu)$. Importantly, note that while choices are made based on perceived values $\tilde{v}$, as shaped by $\mu$, welfare itself is computed on the basis of true values $v$—which are unobserved.

## 3 A Differentiable Proxy for Welfare

We now turn to describing our approach for learning useful decongesting representations.

**Welfare decomposition.** The main difficulty in optimizing Eq. (3) is that we do not have access to true valuations. To remove the reliance on $v$, our first step is to decompose welfare into two terms. Let $W_M = \sum_{ij} \bar{a}_{ij} v_{ij}$ be the expected welfare for a single market $M$, where $\bar{a}_{ij} = \frac{1}{n_j} y_{ij}$ denote expected allocations with $n_j = \sum_i y_{ij}$, and defining $\bar{a}_{ij} = 0$ when $n_j = 0$. We can rewrite $W_M$ as:

$$W_M = \sum_{ij} \frac{1}{n_j} y_{ij} v_{ij} = \sum_j (1 - 1 + \frac{1}{n_j}) \sum_i y_{ij} v_{ij} = \underbrace{\sum_{ij} y_{ij} v_{ij}}_{(I)} + \underbrace{\sum_j (\frac{1}{n_j} - 1) \sum_i y_{ij} v_{ij}}_{(II)} \quad (4)$$

In Eq. (4), term (I) encodes the value users would have gotten from their choices—had there been no supply constraints. Term (II) then corrects for this, and appropriately penalizes excessive allocations.[5]

---

[5]Note that no penalty is incurred if an item is chosen by at most one user, since either $\frac{1}{n_j} - 1 = 0$ or $y_{ij} = 0$.

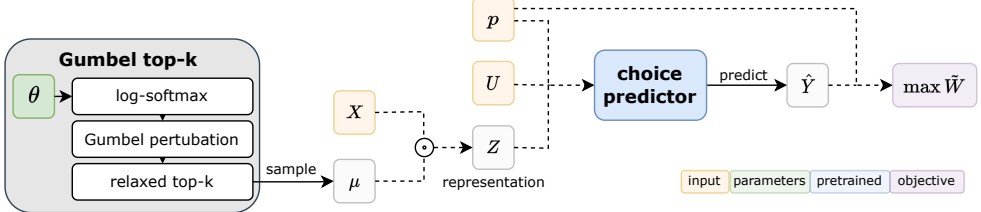

Figure 2: A schematic illustration of our proposed differentiable learning framework.

**Proxy welfare.** Absent the $v_{ij}$, a natural next step is to replace Eq. (4) with a tractable lower bound proxy. For term (I), note that if $y_{ij} = 1$ then $\tilde{v}_{ij} > p_j$ (Eq. (1)), and since $\beta, x \geq 0$, it also holds that $v_{ij} \geq \tilde{v}_{ij}$ (since masking can only decrease perceived value). Hence, we can replace $v_{ij}$ with $p_j$. For term (II), since $\frac{1}{n_j} - 1 \leq 0$, and since we assume $v \leq 1$, using $n_j = \sum_i y_{ij}$ we can write:

$$\widetilde{W}_M = \underbrace{\sum_{ij} y_{ij} p_j}_{= \texttt{selection}(y,p)} - \underbrace{\sum_j \max\{0, n_j - 1\}}_{= \texttt{decongestion}(y)} \leq W_M \tag{5}$$

which removes the explicit dependence on values, and relies only on choices. The two terms in $\widetilde{W}_M$ can now be interpreted as: (I) *selection*, which expresses the total market value of users' choices, as encoded by prices; and (II) *decongestion*, which penalizes excess demand per item. Notice that $n - \texttt{decongestion}(y)$ is simply the number of allocated items, $|\texttt{alloc}(y)|$. To extend beyond unit-supply, we can replace $n_j - 1$ with a more general $n_j - c_j$ when there are $c_j$ copies of item $j$.

Eq. (5) still depends on values implicitly through choices $y$. Our next step is to replace these with *predicted choices*, $\hat{y}_i(\mu) = f(X, p; u_i, \mu)$, where $f$ is a predictive model pretrained on choice data in $\mathcal{S}$:

$$\hat{f} = \operatorname*{argmin}_{f \in F} \sum_{(M,y) \in \mathcal{S}} \sum_{i \in [n]} \mathcal{L}(y_i, f(X, p; u_i, \mu)) \tag{6}$$

for some model class $F$ and loss function $\mathcal{L}$ (e.g., cross-entropy), which decomposes over users. Plugging the learned $f$ into Eq. (5) and averaging over markets in $\mathcal{S}$ obtains our empirical proxy objective:

$$\widetilde{W}_{\mathcal{S}}(\mu) = \frac{1}{N} \sum_{M \in S} \left[ \sum_{ij} \hat{y}_{ij}(\mu) p_j - \sum_j \max\{0, \hat{n}_j(\mu) - 1\} \right] \tag{7}$$

where $\hat{n}_j(\mu) = \sum_i \hat{y}_{ij}(\mu)$. We interpret this as follows: In principle, Eq. (7) seeks representations $\mu$ that entail low congestion by optimizing the `decongestion` term; however, since there can be many decongesting solutions, the additional `selection` term regularizes learning towards good solutions.

**Differentiable proxy welfare.** One challenge in optimizing Eq. (7) is that both predicted choices $\hat{y}$ and masks $\mu$ are discrete objects. To enable end-to-end learning, we replace these with differentiable surrogates. For $\hat{y}$, we substitute 'hard' argmax predictions with 'soft' predictions $\bar{y}_i(\mu)$ using softmax. For masks, instead of optimizing over individual (discrete) masks, we propose to learn masking *distributions*, $\pi_\theta$, that are differentiable in their parameters $\theta$. A natural choice in this case is the multinomial distribution, where $\theta \in \mathbb{R}^d$ assigns weight $\theta_r$ to each feature $r \in [d]$, and masks are constructed by drawing $k$ features sequentially without replacement in proportion to (re)normalized weights, $r \sim \operatorname{softmax}_\tau(\theta)$, where $\tau$ is a temperature hyper-parameter. Our final differentiable proxy objective is:

$$\hat{\theta} = \operatorname{argmax}_{\theta \in \mathbb{R}^d} \widetilde{W}_{\mathcal{S}}(\pi_\theta), \quad \text{where} \quad \widetilde{W}_{\mathcal{S}}(\pi_\theta) = \mathbb{E}_{\mu \sim \pi_\theta}\left[ \widetilde{W}_{\mathcal{S}}(\mu) \right] \tag{8}$$

To solve Eq. (8), we make use of the *Gumbel top-k trick* [29,15]: by reparametrizing $\pi_\theta$, variation in masks due to $\theta$ is replaced with random perturbations $\varepsilon$; this separates $\theta$ from the sampling process, which then permits to pass gradients effectively. We then use the method from [31] to smooth the selection of the top-$k$ elements. For the forward step, the expectation in Eq. (8) is approximated by computing an average over samples $\mu \sim \pi_\theta$. Once $\hat{\theta}$ has been learned, at test time we can either sample from $\pi_{\hat{\theta}}$ as a masking policy, or commit to $\mu_{\hat{\theta}}$, defined to include the $k$ largest entries in $\hat{\theta}$. See Figure 2 for an illustration of the different components of our proposed framework.

**Practical considerations.** One artifact of transitioning from Eq. (4) to Eq. (5) is that the different terms may now become unbalanced in terms of scale. As a remedy, we propose to reweigh them as $(1 - \lambda) \cdot \texttt{selection} + \lambda \cdot \texttt{decongestion}$, where $\lambda$ is a hyper-parameter that can be tuned via experimentation; practically, our empirical analysis suggests that learning is fairly robust to the choice of $\lambda$. In addition, we have also found it useful to add a penalty on non-choices, i.e., $-\sum_i \mathbb{1}\{\hat{y}_i = 0\}$, also weighted by $\lambda$. This can be interpreted as also reducing congestion on the 'no-choice' item, and as accentuating the reward of choosing real items (since no choice gives zero utility; see Appx. G.4).

## 4 THEORETICAL ANALYSIS

The core of our approach relies on minimizing congestion as a proxy to maximizing welfare. It is therefore natural to ask: when does decongestion improve welfare? Focusing on an individual market, in this section we give simple conditions under which allocating more items guarantees an improvement in welfare. Here we consider $p$ to be competitive-equilibrium (CE) prices of the market under full information, meaning that under full information, every item with a strictly positive price is sold and every user can be allocated an item in their demand set. Proofs are deferred to Appendix B.

We start from the strongest type of relation between congestion and welfare, in which allocating more items is always better, irrespective of which items and to which users.

**Definition 1.** *A market with valuations $v_{ij}$ is **congestion monotone** if for all $s \in [m]$, any allocation of $s$ items gives (weakly) better welfare than any allocation of $s' < s$ items.*

Our first result shows that monotonicity holds in economies in which users' valuations for the items are close, as expressed in the following sufficient condition.

**Proposition 1.** *In a market with $n$ users, $m$ items, and valuations $v_{ij}$, denote $v_{\min} = \min_{ij} v_{ij}$ and $v_{\max} = \max_{ij} v_{ij}$. If $\frac{v_{\max} - v_{\min}}{v_{\min}} \leq \frac{1}{m-1}$, then the market is congestion monotone.*

Such monotonicity provides us with very strong guarantees: it will sustain under any user behavior, allocation rule, and randomized outcome. However, this property is demanding in that it considers *all* allocations—whereas some allocations may not be admissible, i.e., result from users choosing on the basis of some representation. We now proceed to pursue this case.

**Definition 2** (Admissible allocation). *An allocation $a$ is **admissible**, denoted $\tilde{a}$, if agents are only assigned their best-response items defined with respect to perceived values $\tilde{v}$ at prices $p$.*

**Definition 3** (Restricted optimality). *An allocation $a$ is **restricted optimal** if $a$ is welfare-optimal at true valuations $v$ in the economy $E = (G_a, N_a)$, where $G_a$ and $N_a$ denote the items and agents, respectively, that are allocated; i.e., the economy restricted to the items and agents that are allocated.*

This property, which in effect defines optimality on a restricted economy, can be established through a set of sufficient conditions by reasoning with suitable notions of competitive equilibrium that arise when working with admissible allocations. To model the way we handle congestion, let $A$ denote a *randomized allocation*, with a product structure defined as follows. Let $G(A)$ denote the set of items allocated.[6] The product structure requires that for each item $j \in G(A)$, some set $N_j$ of agents compete for $j$ with $N_j \cap N_{j'} = \emptyset$, for all $j \neq j'$. Each agent $i \in N_j$ is allocated item $j$ uniformly at random, so that $\Pr_A[i] = 1/|N_j|$ is the probability that $i$ is allocated. We say that a randomized allocation $A$ is admissible if it is a distribution over admissible allocations, and restricted optimal if it is a distribution over restricted optimal allocations. Define $W(A)$ as the expected total welfare at true values, considering the distribution over allocations. We say that a randomized allocation $B$ *extends* $A$ if $G(B) \supset G(A)$ and $\Pr_B[i] \geq \Pr_A[i]$ for all agents $i \in [n]$ (i.e., no agent faces more congestion).

**Theorem 1.** *Given two randomized allocations, $A$ and $B$, where $B$ extends $A$ and $B$ is restricted optimal, then $W(B) \geq W(A)$, with $W(B) > W(A)$ if $v_{ij} > 0$ for all $i, j$.*

The main idea behind this result is that, together with the extension property, and in a way that carefully handles randomization, restricted optimality provides an ordering on welfare.

We now seek conditions under which an admissible allocation is restricted optimal: If these conditions hold for any admissible allocation in the support of a randomized allocation $B$, then by Thm. 1,

---

[6]As explained in Section 2, throughout the paper we consider unique best responses for the users.

$B$ improves welfare relative to all randomized allocations which it extends. We parametrize these conditions by the margin of an admissible allocation, which is defined as follows.

**Definition 4** (Margin). *Let $\tilde{a}$ be an admissible allocation with allocated items and agents $\tilde{G}$ and $\tilde{N}$, resp. Then the **margin** of $\tilde{a}$ is the maximal $\Delta \geq 0$ s.t. $\tilde{v}_{i\tilde{a}_i} - p_{\tilde{a}_i} \geq \max_{j \neq \tilde{a}_i, j \in \tilde{G}}[\tilde{v}_{ij} - p_j] + \Delta, \ \forall i \in \tilde{N}$.*

Denote agent $i$'s *hidden valuation* given mask $\mu$ as $v_{ij}^H = v_{ij} - \tilde{v}_{ij}$.[7] Each of the following conditions is sufficient for restricted optimality and thus the improving welfare claim of Theorem 1:

- **Condition 1: Item heterogeneity is captured in revealed features.** A first property, sufficient for restricted optimality, is that items $\tilde{G}$ allocated in admissible allocation $\tilde{a}$ have similar hidden features, with $|(1-\mu) \odot (x_j - x_{j'})|_1 \leq \Delta, \quad \forall j, j' \in \tilde{G}$, where $\Delta$ is the margin of the admissible allocation, $\mu$ is the mask, and $x_j$ and $x_{j'}$ the features of allocated items $j$ and $j'$, respectively.

- **Condition 2: Agent indifference to hidden features.** A second property is that the agents $\tilde{N}$ allocated in admissible allocation $\tilde{a}$ have relatively low preference intensity for hidden features, with $|(1-\mu) \odot \beta_i|_1 \leq \Delta, \quad \forall i \in \tilde{N}$.

- **Condition 3: Top-item value consistency and low price variation.** A third property relies on the item that is most preferred to an agent considering revealed features also being, approximately, the most preferred considering hidden features. In particular, we require (1) *top-item value consistency*, so that if item $j$ satisfies $\tilde{v}_{ij} \geq \max_{j' \in \tilde{G}} \tilde{v}_{ij'}, \forall i \in \tilde{N}$ (i.e., it is top for $i$ considering revealed features), then $v_{ij}^H + \Delta \geq \max_{j' \in \tilde{G}} v_{ij'}^H$ (i.e., it is approximately top for $i$ considering hidden features); and (2) *small price variation*, so that $|p_j - p_{j'}| \leq \Delta$, for all items $j, j' \in \tilde{G}$.

- **Condition 4: Items have small hidden features.** A fourth property that suffices for restricted optimality is that items have small hidden features, with $|(1-\mu) \odot x_j|_1 \leq \Delta, \quad \forall j \in \tilde{G}$.

- **Condition 5: Agent preference heterogeneity is captured in revealed features.** A fifth property is that the agents $\tilde{N}$ allocated in addmisible allocation $\tilde{a}$ have similar preferences for hidden features, with $|(1-\mu) \odot (\beta_i - \beta_{i'})|_1 \leq \Delta, \quad \forall i, i' \in \tilde{N}$.

## 5 EXPERIMENTS

### 5.1 SYNTHETIC DATA

We first make use of synthetic data to empirically explore our setting and approach. Our main aim is to understand the importance of each step in our construction in Sec. 3. Towards this, here we abstract away optimizational and statistical issues by focusing on small individual markets for which we can enumerate all possible masks, and assuming access to fully accurate predictions $\hat{y}(\mu) = y(\mu)$. The following experiments use $n = m = 8$, $d = 14$, $k = 6$, and CE prices, with results averaged over 10 random instances. Additional results for an alternative decision model can be found in Appendix F.1.

**Variation in preferences.** In general, congestion occurs when users have similar preferences, and our first experiment studies how the degree of preference similarity affects decongestion and welfare. Let $V_{het}, V_{hom} \in \mathbb{R}^{n \times m}$ be value matrices encoding fully-heterogeneous and fully-homogeneous preferences, respectively. We create 'mixture markets' as follows: First, we sample random item features $X$. Then, for each of the above $V_{(i)}$, we extract user preferences $B_{(i)}$ by solving $\min_{B \geq 0} \|BX^\top - V_{(i)}\|_2$. Finally, for $\alpha \in [0, 1]$, we set $B_\alpha = (1 - \alpha)B_{het} + \alpha B_{hom}$ to get $V_\alpha = B_\alpha X^\top$. Thus, by varying $\alpha$, we can control the degree of preference similarity.

Fig. 3 (left) presents welfare obtained by the optimal masks for the following objectives: (i) a welfare oracle (having access to $v$), (ii) a predictive oracle (maximizing $\hat{y}_{ij}(\mu)v_{ij}$ per user), (iii) selection, (iv) decongestion, (v) the welfare lower bound in Eq. (5) (namely selection minus decongestion), and (vi) our proxy objective in Eq. (7). As expected, the general trend is that less heterogeneity entails lower attainable welfare. Prediction and selection, which consider only demand (and not supply) do not fair well, especially for larger $\alpha$. As a general strategy, decongestion appears to be effective; the crux is that there can be many optimally-decongesting solutions—of which some may entail very low welfare (see subplot showing results for all $k$-sized masks in a single market). Of these, our proxy

---

[7]Here we assume w.l.o.g. (given that $0 \leq v \leq 1$) that $\beta \in [0, 1]^m$ and $x \in [0, 1]^m$.

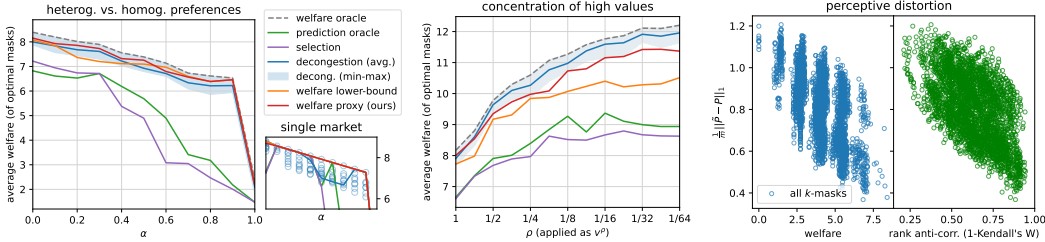

Figure 3: Results on synthetic data. **(Left:)** Welfare obtained by the optimal mask for different objectives, on average and for a single market (inlay). **(Center:)** Performance for increasingly smaller valuation gaps. **(Right:)** Relations between distorted values, welfare, and preference heterogeneity.

objective encourages a decongesting solution that has also high value; results show its performances closely matches the oracle upper bound, despite using $p$ instead of $v$ as in the welfare lower-bound.

**Perceptive distortion.** Partial information can decrease welfare if it causes preferences to shift. This becomes more pronounced if preference shift increases homogeneity, which leads to increased congestion. Since what may cause preferences to shift is the perceptive distortion of values, it would seem plausible to seek representations that minimize distortion. This is demonstrated empirically in Fig. 3 (right). The figure shows evident anti-correlation between perceptive distortion (measured as $\frac{1}{m}\|\tilde{p} - p\|_1$) and welfare across al $k$-sized masks (here we set $\alpha = 0.2$). A similar anti-correlative pattern appears in relation to preference homogeneity from perceived values (measured using Kendall's coefficient of concordance), suggesting that masks are useful if they entail heterogeneous choices.

**Value dispersion.** Although heterogeneity is important, it may not be sufficient. As noted, markets with smaller margins should make our method more susceptible to perceptive distortion. To explore this, we study the effects of 'contracting' the higher-value regime of $v$, achieved by taking powers $\rho < 1$ of $v$ (since $v \in [0, 1]$, we have $v \le v^\rho \le 0$). Fig. 3 (center) shows results for decreasingly smaller powers $\rho$. As expected, since smaller $\rho$ generally increase values, overall potential welfare increases as well. However, as values become 'tighter', this negatively impacts the effectiveness of our approach.

## 5.2 REAL DATA

We now turn to experiments on real data and simulated user behavior. We use two datasets: Movie-Lens, which we present here; and Yelp, which exhibits similar trends, and hence deferred to Appx. G.1.

**Data.** We use the *Movielens-100k* dataset [11], which contains 100,000 movie ratings from 1,000 users and for 1,700 movies, and is publicly-available. Item features $X$ and users preferences $B$ (dimension $d$) were obtained by applying non-negative matrix factorization to the partial rating matrix. User features $U$ (dimension $d'$) were then extracted by additionally factorizing preferences $B$ as $UT^\top \approx B$, where the inferred $T$ can be thought of as an approximate mapping from features to preferences. We experiment in two latent dimension settings: *small* ($d = 12$), which permits computing oracle baselines by enumeration; and *large* ($d = 100$). In both we set $d' = d/2$.

**Setup.** To generate a dataset of markets $\mathcal{S}$, we first sample $m = 20$ items uniformly from $X$, and then sample $L = 240$ sets of $n = 20$ users uniformly from $U$. Masks $\mu$ are sampled according to a 'default' masking policy $\pi_0$ that elicits feature importance from prices, but ensures full support (see 'price predictive' baseline below). For prices $p$ we mainly use CE prices computed per market, but also consider other pricing schemes. Choices $y$ are then simulated as in Eq. (2). Given $\mathcal{S}$, we use a 6-fold split to form different partitions into train test sets. Results are then averaged over 6 random sample sets and 6 splits per sample set (total 36, 95% standard error bars included).

**Method.** For our method of decongestion by representation (DbR), we optimize Eq. (8) using Adam [16] with 0.01 learning rate and for a fixed number of 300 epochs. When $k > d/2$, we have found it useful to set $k \leftarrow d - k$ and learn 'inverted' masks $1 - \mu$. For $\lambda$, our main results use $\lambda = 1 - \frac{k}{2d}$, with the idea that smaller $k$ require more effort placed on decongestion, but note that this very closely matches performance for $\lambda = 0.5$, and that results are fairly robust across $\lambda$ (see Appendix G.3). For $f$ in Eq. (6) we train a bi-linear model (in $u$ and $x$) for 150 epochs using cross-entropy. We consider

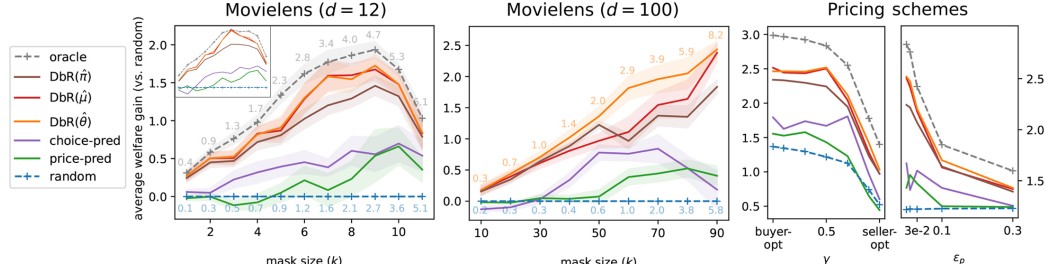

Figure 4: Experiments using real data. **(Left + center:)** Gain in welfare for increasing mask size $k$, for the Movielens dataset using $d = 12$ (left) and $d = 100$ (center) hidden features. Plot lines gain relative to random, numbers show absolute welfare values. **(Right:)** Welfare (absolute) obtained for different prices schemes: (i) buyer- vs. seller-optimal prices, and (ii) increasing additive noise.

three variants of our approach that differ in their test-time usage: (i) $\texttt{DbR}(\hat{\pi})$, which samples masks from the learned policy $\mu \sim \hat{\pi}$; (ii) $\texttt{DbR}(\hat{\mu})$, which commits to a single sampled mask $\hat{\mu} \sim \hat{\pi}$ (having the lowest objective value); and (iii) $\texttt{DbR}(\hat{\theta})$, which constructs and uses a mask $\mu_{\hat{\theta}}$ composed of the top-$k$ entries in the learned $\hat{\theta}$. For further details on implementation and optimization see Appendix E.

**Baselines.** We compare the above to: (iv) $\texttt{price-pred}$, a prediction-based method that uses the top-$k$ most informative features for predicting prices from item features, with the idea that these should also be most informative of values; (v) $\texttt{choice-pred}$, which aims to recover the top-$k$ most important features for users by eliciting an estimate of $T$ (and hence of preferences $\beta$) from the learned choice-prediction model $f$; (vi) an $\texttt{oracle}$ benchmark that optimizes welfare directly (when applicable); and (vii) a $\texttt{random}$ benchmark reporting average performance over randomly-sampled $k$-sized masks.

**Results.** Figure 4 (left, center) shows results for increasing values of $k$. Because overall welfare quickly increases with $k$ for all methods, for an effective comparison across $k$ we plot the relative gain in welfare compared to $\texttt{random}$, with absolute values depicted within. For the $d = 12$ setting (left), results show that our approach is able to learn effective representations attaining welfare that is close to $\texttt{oracle}$. Relative gains increase with $k$ and peak at around $k = 8$. Prediction-based methods generally improve with $k$, but at a low rate. The inlaid plot shows a tight connection to the number of allocated items, suggesting the importance of (de)congestion in promoting welfare (or failing to do so). For $d = 100$ (center), performance of our approach steadily increase with $k$. Here $\texttt{choice-pred}$ preforms reasonably well for $k \approx 50$, but not so for large $k$, nor for small $k$, where $\texttt{price-pred}$ also fails.

**The role of prices.** Because our proxy welfare objective relies on prices for guiding decongestion (for which CE prices are especially useful), we examine the robustness of our approach to differing pricing schemes. Focusing on $d = 12$ and $k = 6$, Figure 4 (right) shows performance for (i) CE prices ranging from buyer-optimal (minimal) to seller-optimal (maximal), and (ii) increasing levels of noise applied to mid-range CE prices. Results show that overall performance degrades as prices become either higher or noisier, demonstrating the general importance of having value-reflective prices. Nonetheless, and despite its reliance on prices, our approach steadily maintains performance relative to others. Appendix G.2 shows similar results for additional variations on pricing schemes.

## 6 DISCUSSION

In this paper, we have initiated the study of decongestion by representation, developing a differentiable learning framework that learns item representations in order to reduce congestion and improve social welfare. Our main claim is that partial information is a necessary aspect of modern online markets, and that systems have both the opportunity and responsibility in choosing representations that serve their users well. We view our approach, which pertains to 'hard' congestion found in tangible-goods markets, and on feature-subset representations, as taking one step towards this. At the same time, 'soft' congestion, which is prevalent in digital-goods markets, also caries many adverse effects. Moreover, there exist various other relevant forms of information representation (e.g., feature ranking, or even other modalities such as images or text). We leave these, as well as the study of more elaborate user choice models, as interesting directions for future work.

ETHICS STATEMENT AND BROADER PERSPECTIVES

Our paper considers the effect of partial information on user choices in the context of online market platforms, and proposes that platforms utilize their control over representations to promote decongestion as a means for improving social welfare. Our point of departure is that partial information is an inherent component of modern choice settings. As consumers, we have come to take this reality for granted. Still, this does not mean that we should take the system-governed decision of what information to convey about items, and how, as a given. Indeed, we believe it is not only in the power of platforms, but also their responsibility, to choose representations with care. Our work suggests that 'default' representations, such as those relying on predictions of user choices, may account for demand—but are inappropriate when supply constraints have concrete implications on user utility.

**Soft congestion.** Although our focus is primarily on tangible-goods, we believe similar arguments hold more broadly in markets for non-tangibles, such as media, software, or other digital goods. While technically such markets are not susceptible to 'hard' congestion since there is no physical limitation on the number of item copies that can be allocated, still there is ample evidence of 'softer' forms of congestion which similarly lend to negative outcomes. For example, digital marketplaces are known to exhibit hyper-popularization, arguably as the product of rich-get-richer dynamics, and which results in strong inequity across suppliers and sellers. Some recent works have considered the negative impact of such soft congestion, but mostly in the context of recommender systems; we believe our conclusions on the role of representations apply also to 'soft' congestion, perhaps in a more subtle form, but nonetheless carrying the same important implications for welfare.

**Limitations.** We consider the task of decongestion by representation in a simplified market setting, including several assumptions on the environment and on user behavior. One key assumption relates to how we model user choice (Sec. 2). While this can perhaps be seen as less restrictive than the standard economic assumption of rationality, our work considers only one form of bounded-rational behavior, whereas in reality there could be many others (our extended experiments in Appendix F.1 take one small step towards considering other behavioral assumptions). In terms of pricing, our theoretical analysis in Sec. 4 relies on equilibrium prices with respect to true buyer preferences, which may not hold in practice. Nonetheless, our experiments in Sec. 5 and Appendix G.2 on varying pricing schemes show that while CE prices are useful for our approach—they are not necessary. Our counterexample in Sec. A suggests that, in the worst case, partially-informed equilibrating prices do not 'solve the problem'. For our experiments in Sec. 5.2, as we state and due to natural limitations, our empirical evaluation is restricted to rely on real data but simulated user behavior. Establishing our conclusions in realistic markets requires human-subject experiments as well as extensive field work. We are hopeful that our current work will serve to encourage these kinds of future endeavours.

**Ethics considerations.** Determining representations has an immediate and direct effect on human behavior, and hence must be done with care and consideration. Similarly to recommendation, decongestion by representation is in essence a policy problem, since committing to some representation at one point in time can affect, through user behavior, future outcomes. Our empirical results in Sec. 5 suggest that learning can work well even when the counterfactual nature of the problem is technically unaccounted for (e.g., training $f$ once at the onset on $\pi_0$, and using it throughout). But this should not be taken to imply that learning of representations in practice can succeed while ignoring counterfactuals. For this, we take inspiration from the field of recommender systems, which despite its historical tendency to focus on predictive aspects of recommendations, has in recent years been placing increasing emphasis on recommendation as a policy problem, and on the implications of this.

While our focus is on 'anonymous' representations, i.e., that are fixed across items and for all users— it is important to note that the *effect* of representations on users is not uniform. This comes naturally from the fact that representations affect the perception of value, which is of course personal. As such, representations are inherently individualized. And while this provides power for improving welfare, it also suggests that care must be taken to avoid discrimination on the basis of induced perceptions; e.g., decongesting by systematically diverting certain groups or individuals from their preferred choices.

Finally, we note that while promoting welfare is our stated goal and underlies the formulation of our learning objective, the general approach we consider can in principal be used to promote other platform objectives. Since these may not necessarily align with user interests, deploying our framework in any real context should be done with integrity and under transparency, to the extent possible, by the platform.

ACKNOWLEDGEMENTS

This research was supported by the Israel Science Foundation (grant No. 278/22), and has received funding from the European Research Council (ERC) under the European Union's Horizon 2020 research and innovation programme (grant agreement No 740282). Gali Noti has also been affiliated with Harvard University and the Hebrew University of Jerusalem during this project. We would like to thank Sophie Hilgard for her conceptual and methodological contributions to the paper in its initial stages.

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

## A  A NOTE ON ADAPTIVE PRICES

Our settings makes the assumption that prices are fixed. This is motivated by settings in which sellers are slow to adapt (or do not adapt at all), and in which representations can be adjusted to take effect more quickly. In this sense, we see representations as adapting to prices—rather than vice versa.

**Adaptive prices.**  An alternative would be to consider prices that adapt to revealed demand, and in particular, prices $\tilde{p}$ that attain competitive equilibrium under perceived values $\tilde{v}$; i.e., "partially-informed competitive equilibrium prices," or "partially-informed prices." These prices would clear the market, but nonetheless have several significant drawbacks:

- First, such prices would completely ignore true valuations $v$, and the actual values that users obtain from items would have no effect on the market. We find this to be unrealistic; a more plausible alternative would be to have (past) true values propagate to influence (future) prices in some manner (e.g., via users posting reviews). Fixed prices can be seen as one (indirect) way to achieve this.

- Second, and relatedly, while partially-informed prices do solve congestion, they do so without any guarantees on welfare; in fact, in our setting, welfare under such prices can be arbitrarily low (see below). This is in contrast to fully-informed prices, which simultaneously minimize congestion *and* maximize welfare.

- Third, in our setting, such partially-informed prices would likely be much lower than prices at full information. This may push sellers to leave the platform if they have an external option, or if prices fall below production costs, this reducing welfare.

- Fourth, and most importantly, partially-informed prices *still depend on what information is revealed*, i.e., they will be different under different masking schemes. Thus, the problem of choosing what information to convey would remain and in fact become more difficult, as learning must now anticipate not only choices, but also induced prices, under possible representations.

Therefore, while learning representations for adapting prices is an intriguing direction, we feel it is deserving of designated future work.

**A constructive example.**  We now show how in our setting, partially-informed prices can give arbitrarily-bad welfare (in the worst case) as a result of their dependence on representations. We prove this by constructing an example in which one representation yields approximately optimal welfare, whereas another yields (approximately) only a small constant fraction, under corresponding partially-informed prices. The construction works by setting half of the features to encode most of the true values of items, and the other half to encode noise. The former subset corresponds to a 'good' representation, for which prices need not adapt much, and hence preserves optimal choices. The latter subset corresponds to a 'bad' representation, which is highly uninformative of values; this causes prices to adapt in a way that entails a 'random' decongested allocation providing very low welfare.

Figure 5 illustrates the values and choices under the different representations and pricing schemes for our example. The precise numerical values used in the example can be found in our code base.

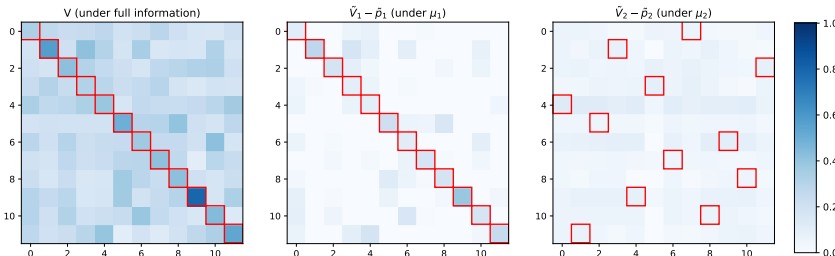

Figure 5: An example market in which different representations entail very different allocations at partial-information market-clearing prices. **(Left:)** The true valuation matrix $V$, with corresponding allocations (red squares) for choices made under full information and CE prices $p^*$. **(Center:)** Perceived values $\tilde{V}_1$ under representation $\mu_1$, minus corresponding partial-information CE prices $\tilde{p}_1$. Resulting allocations are optimal. **(Right:)** Perceived values $\tilde{V}_2$ minus partial-information prices $\tilde{p}_2$ under $\mu_2$. Resulting allocations are highly sub-optimal.

## B    THEORETICAL ANALYSIS

COMPETITIVE EQUILIBRIUM

Let $p = (p_1, \ldots, p_m)$ denote item prices, $a$ denote a feasible allocation (i.e., each item is allocated at most once and each user to at most one item), and $v_i$ agent $i$'s true valuation. Let $\tilde{v}_i$ denote agent $i$'s perceived valuation given mask $\mu$, and $v_{ij}^H = v_{ij} - \tilde{v}_{ij}$ denote agent $i$'s *hidden valuation*. We make the technical assumption that each user has a unique best response, but note the analysis extends to demand sets that are not a singleton by heuristically selecting an item from the demand set.[8]

**Definition 5.** $(a, p)$ *is a competitive equilibrium if (1)* $a_i \in \operatorname{argmax}_j[v_{ij} - p_j, 0]$ *for all* $i$, *and (2) any item with* $p_j > 0$ *is allocated.*

Competitive equilibrium requires that allocation $a$ is (1) a best response for each agent, and (2) maximizes revenue. The following is well known, the proof is included for completeness.

**Theorem 2.** *A CE is welfare optimal.*

*Proof.* The primal assignment problem is

$$\max_a \sum_i \sum_j a_{ij} v_{ij} \tag{9}$$

$$\text{s.t.} \quad \sum_i a_{ij} \leq 1 \quad , \forall j \quad [\text{dual } p_j]$$

$$\sum_j a_{ij} \leq 1 \quad , \forall i \quad [\text{dual } \pi_i]$$

$$x_{ij} \geq 0$$

The dual is

$$\min_{\pi, p} \sum_j p_j + \sum_i \pi_i \tag{10}$$

$$\text{s.t.} \quad \pi_i + p_j \geq v_{ij} \quad \forall i, j \quad [\text{dual } a_{ij}]$$

$$\pi_i, p_j \geq 0.$$

The optimality of CE $(a, p)$, along with $\pi_i = \max_j[v_{ij} - p_j, 0]$ to complete the dual, is established by checking complementary slackness (CS). The primal CS condition is $a_{ij} > 0 \Rightarrow \pi_i + p_j = v_{ij}$, and satisfied since agent $i$ receives an item in its best response set when non-empty (CE), and by the construction of $\pi_i$. The dual CS conditions are $\pi_i > 0 \Rightarrow \sum_j a_{ij} = 1$ and $p_j > 0 \Rightarrow \sum_i a_{ij} = 1$, and satisfied by the CE properties, since every agent with a non-zero demand set gets an item and every item with positive price is allocated. $\qquad \square$

CE prices form a lattice, in general are not unique, and price the *core* of the assignment game [26]. Amongst the set of CE prices, the buyer-optimal and seller-optimal prices are especially salient.

CONGESTION MONOTONICITY

*Proof.* (Proposition 1.): Let $\mathcal{A}_s$ denote the set of all feasible allocations of exactly $s$ items, such that every set $A \in \mathcal{A}_s$ is a set of user-item pairs that represents an allocation of $s$ items. Value matrix $(v_{ij})$ is congestion monotone if and only if for every $s \leq m$ it holds that

$$\max_{A \in \mathcal{A}_{s-1}} \sum_{(i,j) \in A} v_{ij} \leq \min_{A \in \mathcal{A}_s} \sum_{(i,j) \in A} v_{ij}.$$

Next, we define $\delta_{ij} = v_{ij} - v_{min}$ and write every value in $(v_{ij})$ as $v_{ij} = v_{min} + \delta_{ij}$. Using these notations, the congestion monotonicity condition is:

$$v_{min} \geq \left( \max_{A \in \mathcal{A}_{s-1}} \sum_{(i,j) \in A} \delta_{ij} \right) - \left( \min_{A \in \mathcal{A}_s} \sum_{(i,j) \in A} \delta_{ij} \right).$$

---

[8]Alternatively, one can infinitesimally perturb the preference vectors and obtain a unique best response.

Since $s \leq m$ and since the last summation is of positive terms, we have that a sufficient condition is: $v_{min} \geq (m-1) \cdot \max(\delta_{ij}) = (m-1)(v_{max} - v_{min})$, as required. $\qquad\square$

### RESTRICTED OPTIMALITY

We start by discussing deterministic allocations and then proceed to the proof of Theorem 1 and the proofs for the sufficient conditions for restricted optimality. Let *welfare* $W(a) = \sum_i \sum_j a_{ij} v_{ij}$. Let $G_a$ and $N_a$ denote the items and agents, respectively, that are allocated in allocation $a$. Say that $a$ is *restricted optimal* if and only if $a$ is welfare optimal at true valuations $v$ in the economy $E = (G_a, N_a)$; i.e., the economy restricted to the items and agents that are allocated. Say that an allocation $b$ *extends* $a$ if $N_b \supset N_a$ and $G_b \supset G_a$ (i.e., $b$ allocates a strict superset of items and agents).

**Lemma 1.** *Given two allocations, $a$ and $b$, where $b$ extends $a$ and $b$ is restricted optimal, then $W(b) \geq W(a)$, with $W(b) > W(a)$ if $v_{ij} > 0$ for all $i$, all $j$.*

*Proof.* Allocation $a$ is feasible in economy $E_a = (G_a, N_a)$ and thus feasible in economy $E_b = (G_b, N_b)$, and so $W(b) \geq W(a)$ since $b$ is optimal on $E_b$. Moreover, if items have strictly positive value then $W(b') > W(a)$ for allocation $b'$, feasible in $E_b$, that extends $a$ through an arbitrary assignment of items $G_b \setminus G_a$ to $N_b \setminus N_a$. With this, we have $W(b) \geq W(b') > W(a)$, and $b$ strictly improves welfare over $a$. $\qquad\square$

*Proof.* (of Theorem 1.) Consider some deterministic allocation $a$ in the support of $A$, and let $P_1 = \Pr_A[a]$ denote the probability of assignment $a$. Define $P_2 = \sum_{b \in \sup(B), b \text{ extends } a} \Pr_B[b]$, which is the marginal probability of assignments that extend $a$. We have

$$\sum_{b \in \sup(B), b \text{ extends } a} \Pr_B[b] = \sum_{b \in \sup(B), b \text{ extends } a} \prod_{i \in a} \Pr_B[i] \cdot \prod_{i \in b, i \notin a} \Pr_B[i]$$

$$= \prod_{i \in a} \Pr_B[i] \sum_{b \in \sup(B), b \text{ extends } a} \prod_{i \in b, i \notin a} \Pr_B[i] = \prod_{i \in a} \Pr_B[i] \cdot 1 \geq \prod_{i \in a} \Pr_A[i],$$

where the product structure is used to replace the marginalization over the part of the assignment that extends $a$ by probability 1, and the inequality follows since $B$ extends $A$. For any such $b$ that extends $a$, we have $W(b) \geq W(a)$ by Lemma 1, where we use the property that $B$ is restricted optimal and thus each $b$ in the support of $B$ is restricted optimal. Then, since $P_2 \geq P_1$, and considering all such $a$ in the support of $A$, we have $W(B) \geq W(A)$. By considering the case of $v_{ij} > 0$ for all $i$, all $j$, then $W(b) > W(a)$ by Lemma 1, and we have $W(B) > W(A)$. $\qquad\square$

By Theorem 1, to argue that randomized allocation $B$ provides more welfare than randomized allocation $A$ it suffices to argue that (1) each assignment in the support of $B$ is restricted optimal, and (2) $B$ extends $A$ which means that $B$ allocates a superset of the items and each agent is allocated something with at least as much probability in $B$ than $A$ (i.e., no agent faces more congestion).

The first set of conditions, namely Conditions 1, 2, and 3 in the main text, follow from reasoning about the following consistency property, that needs to hold between perceived and true valuations.

**Definition 6** (Pointing consistency.)**.** *An admissible allocation $\tilde{a}$ satisfies* pointing consistency *if, for every agent $i \in \tilde{N}$, the allocated item $\tilde{a}_i$ is the best response of $i$ at true valuations $v_i$.*

In other words, agent $i$ continues to prefer item $\tilde{a}_i$ at prices $p$ when moving from perceived valuation $\tilde{v}_i$ to true valuation $v_i$. The following is immediate.

**Lemma 2.** *Admissible allocation $\tilde{a}$ is restricted optimal if the pointing consistency condition holds.*

*Proof.* $(\tilde{a}, p)$ is a CE (defined with respect to true valuations) in economy $(\tilde{G}, \tilde{N})$. $\qquad\square$

**Lemma 3.** *Admissible allocation $\tilde{a}$ with margin $\Delta$ satisfies pointing consistency (and therefore by Lemma 2 is restricted optimal), when $v_{i\tilde{a}_i}^H \geq v_{ij}^H - \Delta$, for all $j \in \tilde{G}$, all $i \in \tilde{N}$.*

*Proof.* For agent $i$, and any $j \neq \tilde{a}_i$, we have $v_{i\tilde{a}_i} - p_{\tilde{a}_i} = \tilde{v}_{i\tilde{a}_i} - p_{\tilde{a}_i} + v_{i\tilde{a}_i}^H \geq \tilde{v}_{ij} - p_j + \Delta + v_{ij}^H - \Delta = v_{ij} - p_j$, and pointing consistency, where we substitute $\tilde{v}_{i\tilde{a}_i} - p_{\tilde{a}_i} \geq \tilde{v}_{ij} - p_j + \Delta$ (margin condition) and $v_{i\tilde{a}_i}^H \geq v_{ij}^H - \Delta$ (indifference assumption). $\square$

Considering a matrix with agents as rows and items as columns, the property in Lemma 3 is one of "row-dominance" for $\Delta = 0$, such that the value of an agent for its allocated item is weakly larger than that of every other item. For this property, it suffices that there is little variation in the hidden value for any items, which is in turn provided by the set of five conditions.

*Proof.* (of Condition 1) This condition is sufficient for the hidden-value similarity of Lemma 3, since $v_{ij}^H - v_{i\tilde{a}_i}^H = \beta_i^\top (1 - \mu) \odot (x_j - x_{\tilde{a}_i}) = \sum_{k:\mu_k=0} \beta_{ik}(x_{jk} - x_{\tilde{a}_i k}) \leq \sum_{k:\mu_k=0} |\beta_{ik}(x_{jk} - x_{\tilde{a}_i k})| \leq \sum_{k:\mu_k=0} |x_{jk} - x_{\tilde{a}_i k}| = |(1 - \mu) \odot (x_j - x_{\tilde{a}_i})|_1 \leq \Delta$, where the penultimate inequality follows from $0 \leq \beta_{ik} \leq 1$. $\square$

*Proof.* (of Condition 2) This condition is sufficient for the hidden-value similarity of Lemma 3 since $v_{ij}^H - v_{i\tilde{a}_i}^H = \beta_i^\top (1 - \mu) \odot (x_j - x_{\tilde{a}_i}) = \sum_{k:\mu_k=0} \beta_{ik}(x_{jk} - x_{\tilde{a}_i k}) \leq \sum_{k:\mu_k=0} |\beta_{ik}(x_{jk} - x_{\tilde{a}_i k})| \leq \sum_{k:\mu_k=0} |\beta_{ik}| = |(1 - \mu) \odot \beta_i|_1 \leq \Delta$, where the penultimate inequality follows from $0 \leq x_{j'k} \leq 1$, for all item $j'$ and features $k$. $\square$

*Proof.* (of Condition 3) By the margin property, we have $\tilde{v}_{i\tilde{a}_i} - p_{\tilde{a}_i} \geq \tilde{v}_{ij} - p_j + \Delta$, for any $j \in \tilde{G}$, and adding $p_{\tilde{a}_i} \geq p_j - \Delta$ (price variation) we have $\tilde{v}_{i\tilde{z}_i} \geq \tilde{v}_{ij}$, and so $\tilde{a}_i$ is the top item for $i$ given revealed features. Given this, we have $v_{i\tilde{a}_i}^H + \Delta \geq v_{ij}^H$, for all $j \in \tilde{G}$ (top-item value consistency), which is the hidden-value similarity condition of Lemma 3. $\square$

The second set of conditions, namely Conditions 4 and 5 in the main text, come from considering an *approximate column dominance property* on hidden valuations. Considering a matrix with agents as rows and items as columns, column dominance means that the agent to which an item is allocated has weakly larger value for the item than that of any other agent.

**Definition 7** (Approximate column dominance). *An admissible allocation $\tilde{a}$ with margin $\Delta$ satisfies approximate column dominance if, for each item $j \in \tilde{G}$ and agent $i$ allocated item $j$, we have $v_{ij}^H \geq v_{i'j}^H - \Delta$, for all $i' \in \tilde{N}$.*

**Lemma 4.** *Admissible allocation $\tilde{a}$ with margin $\Delta$ is restricted optimal if the approximate column dominance condition holds.*

*Proof.* First, given margin $\Delta$ then

$$\sum_{i \in \tilde{N}} \sum_{j \in \tilde{G}} \tilde{a}_{ij} \tilde{v}_{ij} \geq \sum_{i \in \tilde{N}} \sum_{j \in \tilde{G}} a'_{ij} \tilde{v}_{ij} + |\tilde{G}|\Delta, \quad \text{all } a', \tag{11}$$

since we can reduce $\tilde{v}_{i\tilde{a}_i}$ by $\Delta$ to each agent $i$, leaving the rest of the perceived values unchanged, and this item will still be in the demand set of the agent, and thus $(\tilde{a}, p)$ would be a CE for these adjusted, perceived values (perceived, not true values). Thus, the total perceived value for $\tilde{a}$ is at least $|\tilde{G}| \cdot \Delta$ better than the total perceived value of the next best allocation, considering economy $(\tilde{G}, \tilde{N})$.

Second, we argue that approximate column dominance implies that $\tilde{a}$ approximately optimizes the total hidden value. First, suppose we have exact column dominance, with $v_{ij}^H \geq v_{i'j}^H$, for all $i' \in \tilde{N}$, item $j \in \tilde{G}$, and agent $i$ allocated item $j$. Then, allocation $\tilde{a}$ would maximize hidden values. To see this, consider the transpose of this assignment problem, so that agents become items and items become agents. This maintains the optimal assignment. $\tilde{a}$ is optimal in the transpose economy by considering zero price on each agent and items bidding on agents: by column dominance, each agent is allocated its most preferred agent. By approximate column dominance, we have

$$\sum_{i \in \tilde{N}} \sum_{j \in \tilde{G}} \tilde{a}_{ij} v_{ij}^H + |\tilde{G}|\Delta \geq \sum_{i \in \tilde{N}} \sum_{j \in \tilde{G}} a'_{ij} v_{ij}^H, \quad \text{all } a', \tag{12}$$

and $\tilde{a}$ approximately optimizes total hidden value. This follows by considering the transpose economy, and noting that if we increase $v_{i\tilde{a}_i}^H$ by $\Delta$, to each agent $i$, leaving the other hidden values unchanged,

we have exact column dominance and optimality of $\tilde{a}$. This means that $\tilde{a}$ is at most $|\tilde{G}| \cdot \Delta$ worse than any other allocation. Combining (11) for perceived values and (12) for hidden values, we have

$$\sum_{i \in \tilde{N}} \sum_{j \in \tilde{G}} \tilde{a}_{ij}(\tilde{v}_{ij} + v_{ij}^H) + |\tilde{G}|\Delta \geq \sum_{i \in \tilde{N}} \sum_{j \in \tilde{G}} a'_{ij}(\tilde{v}_{ij} + v_{ij}^H) + |\tilde{G}|\Delta, \quad \text{all } a', \qquad (13)$$

and thus $\tilde{a}$ is restricted optimal, since $\tilde{v}_i j + v_{ij}^H = v_{ij}$. $\qquad\square$

It suffices for approximate column dominance that there is little variation across agents in their hidden value for an item, which is in turn provided by the following properties (approximate column dominance is also achieved by Condition 2).

*Proof.* (of Condition 4) When this condition holds, we have $|v_{ij}^H - v_{i'j}^H| = |\beta_i^\top (1 - \mu) \odot x_j - \beta_{i'}^\top (1 - \mu) \odot x_j| = |(\beta_i - \beta_{i'})^\top (1 - \mu) \odot x_j| = \sum_{k:\mu_k=0} |(\beta_{ik} - \beta_{i'k})x_{jk}| \leq \sum_{k:\mu_k=0} |x_{jk}| = |(1 - \mu) \odot x_j|_1 \leq \Delta$, where the penultimate inequality follows from $0 \leq \beta_{i''k} \leq 1$, for any $i''$, any $k$. This establishes that all pairs of agents have similar hidden value for any given item, and in particular approximate column dominance and $v_{i'j}^H - v_{ij}^H \leq \Delta$ for agent $i$ allocated item $j$ in $\tilde{a}$ and any other agent $i' \in \tilde{N}$. $\qquad\square$

*Proof.* (of Condition 5) With this, we have $|v_{ij}^H - v_{i'j}^H| = |\beta_i^\top (1 - \mu) \odot x_j - \beta_{i'}^\top (1 - \mu) \odot x_j| = |(\beta_i - \beta_{i'})^\top (1 - \mu) \odot x_j| = |\sum_{k:\mu_k=0} (\beta_{ik} - \beta_{i'k})x_{jk}| \leq \sum_{k:\mu_k=0} |(\beta_{ik} - \beta_{i'k})x_{jk}| \leq \sum_{k:\mu_k=0} |\beta_{ik} - \beta_{i'k}| = |(1 - \mu) \odot (\beta_i - \beta_{i'})|_1 \leq \Delta$, where the penultimate inequality follows from $0 \leq x_{jk} \leq 1$ for any item $j$. This establishes that all pairs of agents have similar hidden value for any given item, and in particular approximate column dominance and $v_{i'j}^H - v_{ij}^H \leq \Delta$ for agent $i$ allocated item $j$ in $\tilde{a}$ and any other agent $i' \in \tilde{N}$. $\qquad\square$

## C  METHOD: ADDITIONAL DETAILS

Although our approach makes use of prediction, in essence, the problem of finding optimal representations is counterfactual in nature. This is because choosing a good mask requires anticipating what users *would have chosen* had they made choices under this new mask; these may differ from the choices made in the observed data. As such, decongestion by representation is a policy problem. This has two implications: on how data is collected, and on how to predict well.

### C.1  DEFAULT POLICY

To facilitate learning, we assume that training data is collected under representations determined according to a 'default' stochastic masking policy, $\pi_0$. The degree to which we can expect data to be useful for learning counterfactual masks depends on how informative $\pi_0$ of other representations. In particular, if there is sufficient variation in masks generated by $\pi_0$, then in principle it should be possible to generalize well from $\pi_0$ to a learned masking policy, $\hat{\pi}$ (which can be deterministic). We imagine $\pi_0$ as concentrated around some reasonable default choice of mask, e.g., as elicited from a predictive model, or which includes features estimated to be most informative of user values. However, $\pi_0$ must include some degree of randomization; in particular, to enable learning, we require $\pi_0$ to have full support over all masks, i.e., have $P_{\pi_0}(\mu) \geq \epsilon$ for all $\mu$ and for some $\epsilon > 0$. In our experiments we set $\pi_0$ to have most probability mass concentrated around features coming from a predictive baseline (e.g., `price-pred`), but with some probability mass assigned to other features.

### C.2  COUNTERFACTUAL PREDICTION

Representation learning is counterfactual since choices at test time depend on the learned mask. At train time, counterfactuality manifests in predictions: for any given $\mu$ examined during training, our objective must emulate choices $y(\mu)$, which rely on $v$, via predictions $\hat{y}(\mu)$, which rely only on observed features $u, X$ and prices $p$. As such, we must make use of choice data sampled from $\pi_0$ to predict choices to be made under differing $\mu$. There is extensive literature on learning under distribution shift, and in principle any method for off-policy learning should be applicable to our

case. One prominent approach relies on *inverse propensity weights*, which weight examples in the predictive learning objective according to the ratio of train- to test-probabilities,

$$w_\pi(\mu) = \frac{P_\pi(\mu)}{P_{\pi_0}(\mu)}$$

for all masks $\mu$ in the training data, which are then used to modify Eq. (6) into:

$$\hat{f} = \operatorname*{argmin}_{f \in F} \sum_{(M,y) \in \mathcal{S}} \sum_{i \in [n]} w_\pi(\mu) \mathcal{L}(y_i, f(X, p; u_i, \mu)) \tag{14}$$

For the default policy, propensities $\rho = P_{\pi_0}(\mu)$ are assumed to be collected and accessible as part of the training set. For the current policy $\pi$, $P_\pi(\mu)$ can be approximated from the Multivariate Wallenius' Noncentral Hypergeometric Distribution, which describes the distribution of sampling without replacement from a non-uniform multinomial distribution. . This makes the predictive objective unbiased with respect to the shifted target distribution, and as a result, makes Eq. (8) appropriate for the current $\pi$.

In our case, because the shifted distributions are not set a-priori, but rather, are determined by the learned representations themselves, our problem is in fact one of *decision-dependent distribution shift*. Our proposed solution to this is to alternate between: (i) optimizing $f_t$ in Eq. (6) to predict well for data corresponding to the current mask $\mu_{t-1}$, holding parameters $\theta_{t-1}$ fixed; and (ii) optimizing $\mu_t$ by updating parameters $\theta_t$ in Eq. (8) for a fixed $f_t$. That is, we alternate between training the predictor on a fixed choice distribution, and optimizing representations for a fixed choice predictor.

Nonetheless, in our experiments we have found that simply training $f$ to predict well on $\pi_0$—without any reweighing or adjustments—obtained good overall performance, despite an observed reduction in the predictive performance of $f$ on counterfactual choices made under the learned $\mu$ (relative to predictive performance on $\pi_0$).

## D  EXPERIMENTAL DETAILS: SYNTHETIC

Experiments were implemented in Python. See supplementary material for code.

**Prices.**  For computing CE prices we used the cvxpy convex optimization package to implement Eq. (10). This give *some* price vector in the core. To interpolate between buyer-optimal and seller-optimal core prices, we adjust Eq. (10) by: (i) solving the original Eq. (10) to obtain the optimal dual objective value; (ii) adding a constraint for the objective value to obtain the optimal value; and (iii) modifying the current objective to either minimize prices (for buyer-optimal) or maximize prices (for seller-optimal).

**Preferences.**  To generate mixture value matrices, we first sample two random item features matrices $X_{(1)}, X_{(2)} \in [0,1]^{n \times d}$ with entries sampled independently from the uniform distribution over $[0,1]$. Next, we generate a fully-heterogeneous value matrix $V_{(1)} = V_{\text{het}}$, and a fully-homogeneous matrix $V_{(2)} = V_{\text{hom}}$. The heterogeneous matrix is constructed by taking the preference vector $(m, m-1, \dots, 1)$, normalized to $[0,1]$, and creating a circulant matrix, so that user $i$ most prefers item $i$, and then preferences decreasing in items with increasing indices (modulo $m$). The homogeneous matrix is constructed by assigning the same preference vector to all users.[9] Finally, to obtain the corresponding $B_{(i)}$, we solve for the convex objective $\min_{B \geq 0} \|BX^\top - V_{(i)}\|_2$, and for $\alpha \in [0,1]$, set $B_\alpha = (1-\alpha)B_{\text{het}} + \alpha B_{\text{hom}}$ and $X = X_{(1)} + X_{(2)}$, which gives the desired $V_\alpha = B_\alpha X^\top$.

**Optimization.**  Because we consider small $n, m, d$, and because as designers of the experiment we have access to $v$, in this experiment we are able to compute measures that rely on $v$. In particular, by enumerating over all $d$-choose-$k$ possible masks, we are able to exactly optimize the considered objectives, compute the welfare oracle upper bound, and obtain all optimal solutions in case of ties (as in the case of the decongestion objective).

---

[9]We also experimented with adding noise to each $V_{(i)}$ (small enough to retain preferences), but did not observe this to have any significant impact on results.

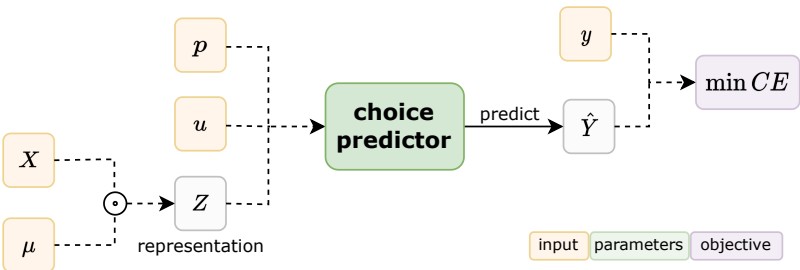

Figure 6: A schematic illustration of our choice prediction model.

# E  EXPERIMENTAL DETAILS: REAL DATA

## E.1  DATA GENERATION

**Data and preprocessing.**  The Movielens 100k dataset is available at https://grouplens.org/da tasets/movielens/100k/. NMF on partial rating matrix was done by *surprise*[10] python package For Movielens, as rating vlues range from 1 to 5, we normalize then into $[0, 1]$ by dividing the user preferences matrix $B$ by a factor of 5.

**Prices.**  CE prices $p^*$ were computed by solving the dual LP in Eq. (10), similarly to the synthetic experiments. For varying prices between buyer-optimal ($p_{\text{buyer}}$) and seller-optimal ($p_{\text{seller}}$) CE prices, we interpolate between $p_{\text{buyer}}$ and $p^*$ for $\gamma \in [0, 0.5]$, and between $p^*$ and $p_{\text{seller}}$ for $\gamma \in (0.5, 1]$, this since interpolating directly between $p_{\text{buyer}}$ and $p_{\text{seller}}$ is prone to exhibiting many within-user ties as an artifact, and since $p^*$ is often very close to the average price point $(p_{\text{buyer}} + p_{\text{seller}})/2$.

**Default masking policy.**  As discussed in Appendix C.1, our method requires training data to be based on masks generated from a default masking policy, $\pi_0$. We defined $\pi_0$ to be concentrated around the features selected by the `price-pred` predictive baseline, but ensure all features have strictly positive probability. In particular, let $\mu_0$ be the mask including the set of $k$ features as chosen by `price-pred`. Then $\theta$ for $\pi_0$ is constructed as follows: first, we assign $\theta_i = 1$ for all $i \notin \mu_0$; then, we assign $\theta_i = 3$ for all $i \in \mu_0$; finally, we normalize $\theta$ using a softmax with temperature 0.05, this resulting in a distribution over features that strictly positive everywhere but at the same time tightly concentrated around $\mu_0$, and in a way which depends on $k$ (since different $k$ lead to different normalizations). An example $\pi_0$ is shown in Fig 7 (left).

## E.2  OUR FRAMEWORK

**Choice prediction.**  The choice prediction model $f$ is trained to predict choices (including null choices) from training data. For the class of predictors $F = \{f\}$, we use item-wise score-based bilinear classifiers parameterized by $W \in \mathbb{R}^{d \times d'}$, namely:

$$f_W(X, p; u, \mu) = \underset{x \in X}{\text{argmax}}\, u^\top W(x \odot \mu) - p$$

There are implemented as a single dense linear layer, and for training, the argmax is replaced with a differentiable softmax. We found learning to be well-behaved even under low softmax temperatures, and hence use $\tau_f = 5\text{e}{-}4$ throughout. For training we used cross entropy loss as the objective. For optimization we Adam for 150 epochs, with learning rate of $1\text{e}{-}3$ and batch size of 20. See Figure 6 for a schematic illustration. Training data used to train $f$ includes user choices $y$ made on the basis masks $\mu$ sampled from the default policy, $\mu \sim \pi_0$. Nonetheless, as described in C.2, recall that we would like $f$ to predict well on the final learned mask $\mu$, but also on other masks encountered during training, and more broadly—on any possible mask. Figure 7 (center+right) shows, for $d = 12$ and $d = 100$ and as a function of $k$, the accuracy of $f$ on (i) data representative of the training distribution (i.e., masks sampled from $\pi_0$), and (ii) data which includes masks sampled uniformly at random from the set of all possible $k$-sized masks. As can be seen, across all $k$, performance on arbitrary masks

---
[10]https://surpriselib.com/

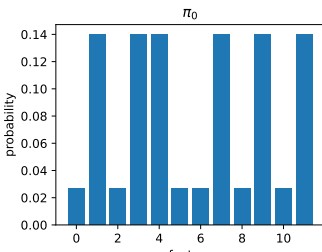 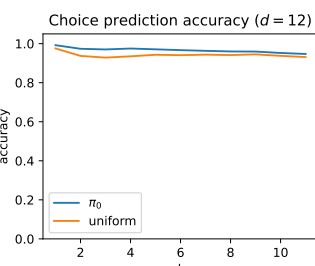 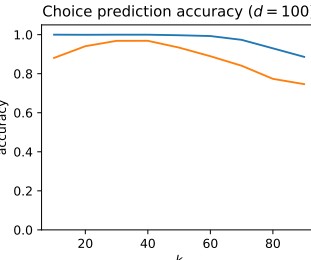

Figure 7: **(Left):** An example for default policy $\pi_0$ ($d = 12, k = 6$). Each feature (in x-axis) is assigned with a probability (y-axis) to be drawn from the categorical distribution. High probability is assigned to predictive features from `price-pred`. **(Center+right):** Accuracy of our choice predictor $f$, for Movielens with $d = 12$ (center) and $d = 100$ (right).

closely matches in-distribution performance for $d = 12$, and remains relatively high for $d = 100$ (vs. random performance at 5% for $m = 20$).

**Representation learning.** The full-framework model consists of a Gumbel-top-$k$ layer, applied on top of a 'frozen' choice prediction model $f$, pre-trained as described above. The Gumbel-top-$k$ layer has $d$ trainable parameters $\theta \in \Theta = \mathbb{R}^d$; once passed through an additional softmax layer, this constitutes a distribution over features. As described in the main paper, given this distribution, we generate random masks by independently sampling $k$ features $i \sim \theta_i$ without replacement (and re-normalizing $\theta$). However, to ensure our framework is differentiable, we use a *relaxed-top-$k$* procedure for generating 'soft' $k$-sized masks, and for each batch, we sample in this way $N$ soft masks, for which we adopt the procedure of [31].

Given a sampled batch of masks $\{\mu\}$, these are then plugged in to the prediction model $f$ to obtain $\hat{y}(\mu)$, and finally our proxy-loss $-\tilde{W}$ is computed. Optimization was carried out using the Adam optimizer for 300 epochs (at which learning converged for most cases) and with a learning rate of $1e{-}2$. We set $N = 20$, and use temperatures $\tau_{\text{Gumbel}} = 2$ for the Gumbel softmax, $\tau_{\text{top-}k} = 0.2$ for the relaxed top-$k$, and $\tau_f = 0.01$ for the softmax in the pre-trained predictive model $f$. Since the selection of the top-$k$ features admits several relaxations, for larger $k > d/2$, we have found it useful to instead consider $k \leftarrow d - k$ in learning, and then correspondingly use 'inverted' masks $\mu \leftarrow 1 - \mu$.

**Variants.** As noted, we evaluate three variants of our approach that differ in their usage at test-time:

- `DbR`($\hat{\theta}$): Constructs a mask from the top-$k$ entries in the learned $\hat{\theta}$.

- `DbR`($\hat{\mu}$): A heuristic for choosing a mask on the basis of training data. Here we sample 20 masks $\hat{\mu}$ according to the multinomial distribution defined by the learned $\hat{\theta}$, and commit to the sampled mask obtaining the lowest value on the proxy objective.

- `DbR`($\hat{\pi}$): Emulates using $\hat{\theta}$ as a masking policy $\hat{\pi} = \pi_{\hat{\theta}}$. Here we sample 50 masks $\mu \sim \hat{\pi}$, evaluate for each sampled mask its performance on the entire test set, and average.

### E.3 BASELINES

- Price predictive (`price-pred`): Selects the $k$ most informative features for the regression task of predicting the price of items, based on item its features. Data includes features and prices for all items that appear in the dataset (recall markets include the same set of items, but items can be priced differently per market). We use the Lasso path (implementation by *scikit-learn*[11]) to order features in terms of their importance for prediction, and take as a mask the top $k$ features in that order.

- Choice predictive (`choice-pred`): Selects the $k$ most informative features for the classification task of predicting user choices from user and item features. For this baseline we use the predictive model $f$, where we interpret learned weights $W = \hat{T}$ as an estimated of the true underlying mapping $T$ between user features $u$ and (unobserved) preferences $\beta$. Inferred

---

[11]https://scikit-learn.org/stable/modules/generated/sklearn.linear_model.lasso_path

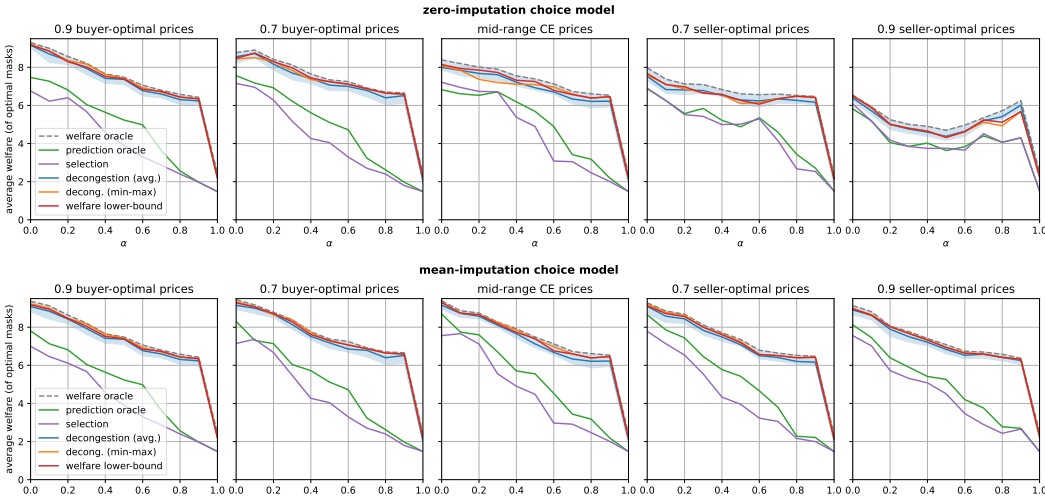

Figure 8: A replication of results on synthetic data (Sec. 5.1) on an alternative choice model based on mean-imputed values (top), in comparison to the zero-imputed choice model studied in the main paper (bottom). Evaluation is also extended to additional CE price schemes.

parameters $\hat{T}$ are then used to obtain estimated preferences per user via $\hat{\beta} = u\hat{T}$. We then average preferences over users, to obtain preferences representative of an 'average' user, and from which we take the top $k$-features, we we interpret as accounting for the largest proportion of value.

- Random (`random`): Here we report performance averaged over 100 random masks sampled uniformly from the set of all $k$-sized masks.

### E.4 IMPLEMENTATION

**Code.** All code is written in python. All methods and baselines are implemented and trained with Tensorflow[12] 2.11 and using Keras. CE prices were computed using the convex programming package cvxpy[13].

**Hardware.** All experiments were run on a Linux machine wih AMD EPYC 7713 64-Core processors. For speedup runs were parallelized each across 4 CPUs.

**Runtime.** Runtime for a single experimental instance of the entire pipeline was clocked as:

- $\approx 6.5$ minutes for the $d = 12$ setting
- $\approx 13.5$ minutes for the $d = 100$ setting

Data creation was employed once at the onset.

## F ADDITIONAL EXPERIMENTAL RESULTS: SYNTHETIC DATA

### F.1 MEAN-IMPUTATION CHOICE MODEL

In this section we replicate our main synthetic experiment in Sec. 5.1 on a different user choice model. In the main part of the paper, we model users as contending with the partial information depicted in representations by assuming that unobserved features do not contribute towards the item's value.

In particular, here we consider users who replace masked features with *mean-imputed values*: for example, if some feature $\ell$ is masked, then features $x_{j\ell}$ are replaced with the 'average' feature, $\bar{x}_\ell =$

---

[12] https://www.tensorflow.org/
[13] https://www.cvxpy.org/

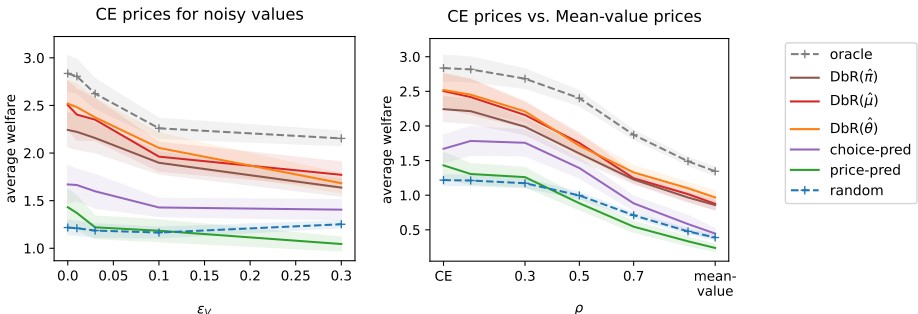

Figure 9: Other prices schemes for real data experiments. **(Left:)** Welfare (absolute) obtained for CE prices computed on noisy valuations $v + \epsilon_v$ for increasing additive noise $\epsilon_v$. **(Right:)** Welfare (absolute) obtained for prices that interpolate between CE prices and heuristic (non-CE) prices set to average user values.

$\frac{1}{m} \sum_{j'} x_{j'\ell}$, computed over and assigned to all market items $j$. This is in contrast to the choice model defined in (see Sec. 2) which relies on zero-imputed values. The main difference is that with mean imputation, (i) perceived values can also be *higher* than true values (e.g., if $\bar{x}_\ell > x_{j\ell}$ for some item $j$); and (ii) our proxy welfare objective in Eq. (5) is no longer a lower bound on true welfare. Nonetheless, we conjecture that if mean-imputed perceived values do not dramatically distort inherent true values, then proxy welfare can still be expected to perform well as an approximation of true welfare.

Figure 8 shows performance for all methods considered in Sec. 5.1 on mean-imputed choice behavior, for increasing $k$ and for a range of possible CE prices. For comparison we also include results for our main zero-imputed choice model (mid-range CE prices are used in Sec. 5.1). As can be seen, our approach retains performance for mean-imputed choices across all considered pricing schemes. Whereas for zero-imputed choices overall welfare decreases when prices are higher (likely since higher prices increase null choices), mean-imputed choices exhibit a similar degree of welfare regardless of the particular price range.

## G  ADDITIONAL EXPERIMENTAL RESULTS: REAL DATA

### G.1  ADDITIONAL DATASET - YELP RESTAURANTS

Here we present a replication of our main experiment in and an additional dataset—the restaurants portion of the Yelp! reviews dataset, which is publicly-available.[14] We use the same preprocessing procedure and experimental setup as for MovieLens (see Appendix E), but also filter to keep restaurants that received at least 20 reviews, and users that gave at least 20 reviews. Figure 10 shows results.

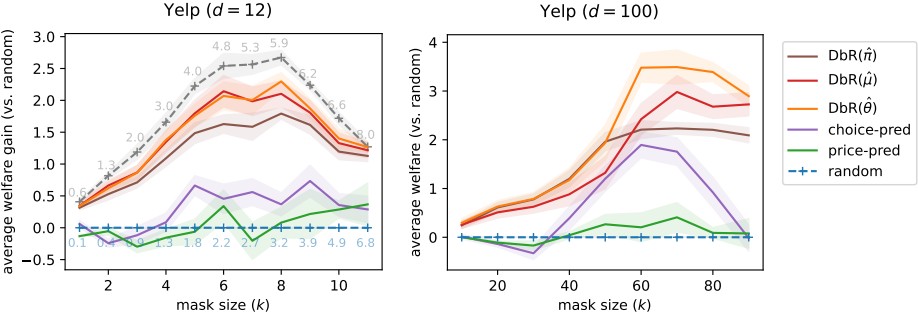

Figure 10: A replication of our main experiment on the Yelp restaurant reviews dataset.

---

[14]https://www.kaggle.com/datasets/yelp-dataset/yelp-dataset

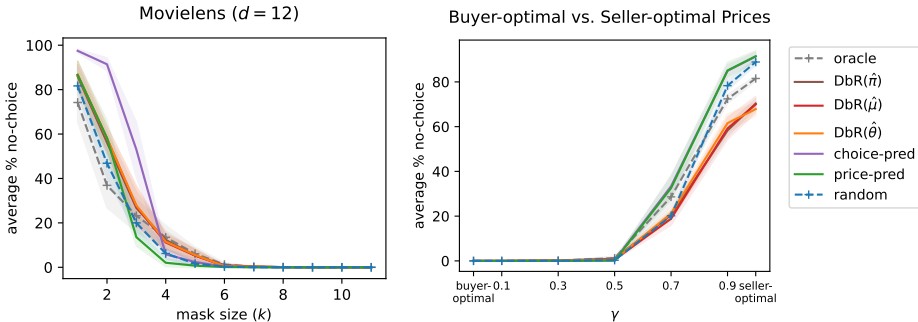

Figure 11: Null-item choices (equivalent to no-choice). **(Left:)** Main experiment on $d = 12$. **(Right:)** Prices scheme in which CE prices ranges from buyer-optimal (minimal) to seller-optimal (maximal).

As can be seen, general trends are qualitatively similar to the those observed for the Movielens dataset (Figure 4 in the main paper).

### G.2 PRICING SCHEMES

Here we present robustness results for additional pricing schemes, which complement our results from Sec. 5.2. In particular, we examined performance for:

- Prices set by solving Eq. (10), but for noisy valuations $v + \epsilon_v$, for increasing levels of noise $\epsilon$. These simulate a setting where prices are CE, but for the 'wrong' valuations, $p^*(v + \epsilon_v)$. Results are shown in Figure 9 (left).

- Prices that interpolate from mid-range CE prices (as in the main experiments) to heuristically-set, non-CE prices. Specifically, here we use prices based on average values assigned by users to items. Results are shown in Figure 9 (right).

Overall, as in the main paper, moving away from CE prices causes a reduction in potential welfare, and in the performance of all methods. Results here demonstrate that in the above additional pricing settings, our approach is still robust in that it maintains it's relative performance compared to baselines and the welfare oracle.

### G.3 THE IMPORTANCE OF $\lambda$

In principle, and due to the counterfactual nature of learning representations (see Appendix C, tuning $\lambda$ requires experimentation, i.e., deploying a learned masking model $\hat{\mu}$ trained on data using some $\lambda$, to be evaluated on other candidate $\lambda'$. Nonetheless, in our experiments we observe that learning is fairly robust to the choice of $\lambda$, even if kept constant throughout training. Figure 14 (bottom-right) shows welfare (normalized) obtained for a different $\lambda$ on Movielens using $d = 12$. As can be seen, any $\lambda > 0.5$ works well and on par with our heuristic choice of $\lambda = 1 - k/2d$, used in Sec. 5.

### G.4 NO-CHOICE PENALTY

One empirical observation that came up during experimentation was that the optimization of our proposed differential welfare proxy occasionally converged to a degenerate solution in which all users choose the null option. To circumvent this, we added to the objective a penalty term that discourages the outcome in which all users choose the null item (see end of Sec. 3). This proved useful in steering the optimization trajectory away from such undesired local optima, which trivially implies no congestion but for the "wrong" reasons, and is by definition sub-optimal (in terms of both the proxy objective value and actual welfare). A possible concern that could arise from using this penalty would be that it could inadvertently coerce users to always choose some (non-null) item, which is undesirable. However, Figure 11 (Left) demonstrates that this is not the case: user do indeed choose the null item even when the penalty is present in the objective. Furthermore, and as ban be expected, such non-choices become even more frequent when prices are higher.

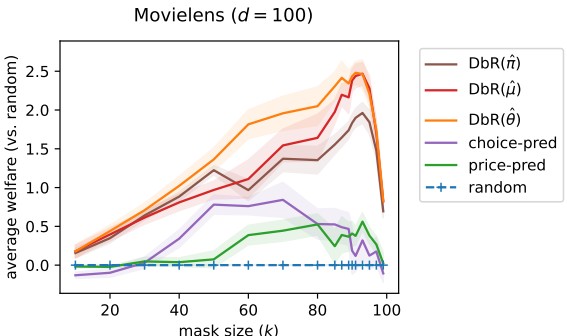

Figure 12: Results on Movielens with $d = 100$ for additional values of $k > 90$.

## G.5 HIGHER-RESOLUTION RESULTS FOR MOVIELENS

When examining Fig. 4 from Sec. 5.2, which shows results for the Movielens dataset, it may seem as if there is a qualitative difference between outcomes for $d = 12$ and $d = 100$: whereas for $d = 12$ the improvement of the DbR methods in terms of welfare (relative to random) seems to increase with $k$ and then decrease, for $d = 100$, it appears to be only increasing. This, however, is an artifact of the range of values of $k$ considered in each experiment; Clearly, for any $d$, performance cannot only increase in $k$, since for $k = d$ performance for all methods is the same, and so the relative gain vs. the random baseline is always zero (as it is also for $k = 0$). Hence, and whereas for $d = 12$ performance peaks at around $k = 9$, the optimal point for $d = 100$ (again, in terms of relative welfare gain) is for some $k \in [90, 100]$.

To validate this, we evaluated performance on a tighter grid of values for large $k$, and in particular for $k \in \{91, 92, \dots, 99\}$. Results are shown in Fig. 12, together with all previous $k$. As expected, for $d = 100$ relative welfare gains do indeed increase first and then decrease, with the maximum attained at around $k = 96$.

## G.6 RELATIVE AND ABSOLUTE PERFORMANCE

In Sec. 5, for our experiments which vary $k$, we chose to portray results normalized from below to match random performance (random). This was mainly since the overall effect on performance of increasing $k$ is larger than that which can be obtained by any method (i.e., the gap between random and oracle). For completeness, Figure 14 (top row) shows unnormalized results, which show in absolute terms how overall performance increases for $k$. Figure 14 (bottom-left) shows results normalized from both below (matching random) and above (matching oracle); as can be seen, our approach obtains fairly constant relative performance across $k$. For completeness, Figure 13 shows in more detail the number of allocated items for the $d = 12$ setting.

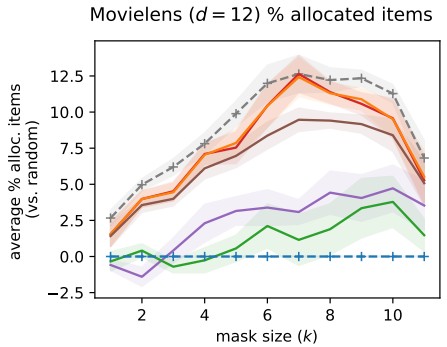

Figure 13: Number of unique items (enlarged version of inlay in Fig. 4 (left)).

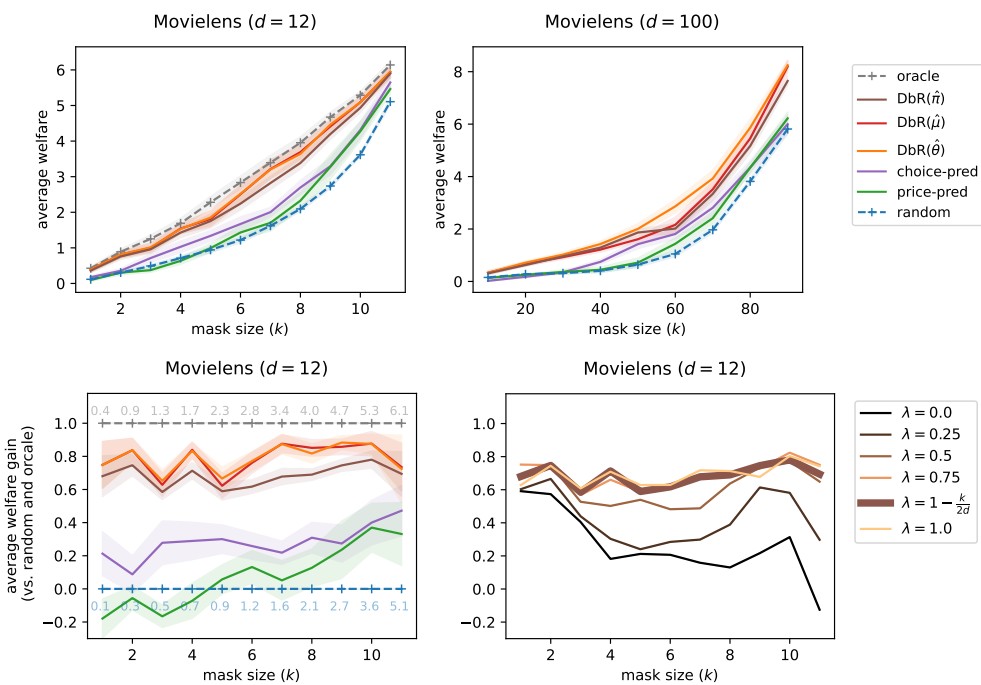

Figure 14: **(Top row:)** Unnormalized results, showing how potential and obtained welfare increase with $k$. **(Bottom-left:)** Results normalized to 1 from above (matching the `oracle`, as in the main paper) and below to 0 (matching `random`). Our approach shows relative performance that is fairly constant across $k$. **(Bottom-right:)** Performance (also normalized) for various fixed $\lambda$.

