# OpenReview forum: "Decongestion by Representation: Learning to Improve Economic Welfare in Marketplaces"
_ICLR.cc/2024/Conference — ICLR 2024 poster_

### Official Review · Reviewer_1Wxo · 2023-10-21

**Soundness:** 4 excellent
**Presentation:** 3 good
**Contribution:** 3 good
**Rating:** 6
**Confidence:** 3

**Summary:**

Online platforms typically sell goods in a decentralized fashion which complicates the equilibration of supply and demand as sellers set prices and buyers must make decisions under imperfect information. Platforms can typically only control the set of information provided to buyers. In that sense platforms want to find the representations (or sets of information to present) that improves social welfare. The paper in particular tackles the challenge of congestion where supply does not meet demand. The authors develop a learning technique that to find representations that reduce congestion which they argue using equilibrium analysis improves social welfare under a partial information framework.

**Strengths:**

The paper studies an important question in online platforms of what information should be presented in order to maximize social welfare. The paper poses this question through a nice framework of learning representations which allows us to employ machine learning tools to optimize this objective.  The modeling framework gives an interesting perspective to design of online platforms and has the potential for practical contributions. Furthermore, the paper uses both simulated and real world data to underscore it's point which is helpful in seeing the contributions of the paper.

**Weaknesses:**

- Solving the objective requires smoothing a discrete object. As the authors note this can add some practical difficulties. One approach they take is to penalize the "no-choice" option. Is this reasonable in practice? Often the "outside" option can have a large market share in studies of demand.

- The theoretical analysis feels a little misplaced. Specifically, the theoretical analysis seems to focus on the perfect information equilibrium, is this really relevant in the online marketplace scenario as the authors note earlier in the paper? Furthermore, this focuses on proxying social welfare with reducing congestion but this may not be the objective platforms want to optimize for. In fact, decongestion seems practical enough objective on its own. This also involves making assumptions like "item heterogeneity is captured in revealed features" which is likely violated in practice.

- It seems like the results in the real data section may rely somewhat on how choices are simulated through prices. Can you comment on how this could affect results if there is some additional dependencies in choices.

**Questions:**

How easily can this optimization framework be generalized to optimize other objectives of interest? i.e. maximizing platform profits

Can you characterize further some of the loss due to smoothing the discrete problem?

Is the oracle benchmark based on the optimal representation?

Additional questions posed in weaknesses.

---

> ### Author Response · Authors · 2023-11-19
> **Response (part 1/2)**
>
> Thank you for your careful reading and useful comments. We appreciate your reviewing efforts, and hope our response below helps in addressing your questions.
>
> **Is penalizing the "no-choice" option reasonable in practice?**
> The no-choice penalty was added for optimization purposes - not to drive users away from this option. During experimentation, we noticed that the optimization occasionally converged to a degenerate solution in which all users choose the null option. The no-choice penalty is used to steer optimization away from this local maxima, which trivially implies no congestion but for the “wrong” reasons, and is suboptimal (in terms of both the proxy objective value and actual welfare). In the experiments we observed that despite the penalty users do select the null item, and quite frequently so when prices are higher, which is expected. We will add these results to the final version.
>
>
> **”The theoretical analysis seems to focus on the perfect information equilibrium, is this really relevant in the online marketplace scenario as the authors note earlier in the paper?”**
> Please note that CE prices do not imply “perfect” (or complete) information, and the theory of market equilibrium is generally silent as to how equilibrium prices arise. Rather, this is conceptualized as an adjustment (or `tatonnement’) process whereby prices settle down to values wherein supply balances demand or with sellers becoming informed through participation in other markets. By these means, equilibrium prices emerge - irrespective of informational considerations. In our settings, we emphasize the fact that the learning platform *does not* observe consumer values; in fact, this is precisely the reason why prices are useful within our proxy objective, providing an alternative to these unobserved values.
>
> **“Decongestion seems practical enough objective on its own.”**
> Please note that in general, decongestion is in itself insufficient: any solution in which users choose differently - or do not choose at all - leads to zero congestion, but can be arbitrarily bad. The notion of monotonicity (page 6) provides a sufficient condition for when increasing the number of allocations (e.g., by decongestion) also improves welfare.
>
>
> **Results on real data may rely on how choices are simulated through prices. How would this affect results under additional dependencies in choices?**
> If you mean dependencies in choices as reflected by pricing (e.g., through the market mechanism), then please see the extended results on alternative pricing schemes in Appendix G.2 (in addition to those presented in Sec. 5). If you mean alternative choice models, then please see the results we give for an alternative choice model in Appendix F.1.
>
>
> **”How easily can this optimization framework be generalized to optimize other objectives of interest? i.e. maximizing platform profits.”**
> The construction of the proxy objective itself (Eq. (5)) is specific to welfare, but the technical aspects of our framework which allow to pass gradients through masks are quite general, and should be compatible with other objectives. Revenue is one likely example, as are combinations of revenue and welfare, but such extensions remain outside the focus of the present paper. At the same time, it is important to emphasize that any changes to the objective should be subject to careful scrutiny as to possible social or ethical implications.
>
>
>
> **”Is the oracle benchmark based on the optimal representation?”**
>
> Yes - the oracle chooses the discrete mask that optimizes true welfare; this solution is obtained (for small d) via enumeration.

---

> > ### Author Response · Authors · 2023-11-19
> > **Response (part 2/2)**
> >
> > **”Can you characterize further some of the loss due to smoothing the discrete problem?”**
> > There are three smoothing elements in our proxy:
> > 1. Replacing discrete masks $\mu$ with distributions $\pi$:
> > We consider this to be the primary smoothing element. The potential loss here is due to adding variance around a focal $\mu$ (e.g., the mean), which places more weight on other possibly sub-optimal masks; one indication of this is the differences in test performance across DbR variants (two of which extract a different discrete $\mu$ from the same learned $\hat{\pi}$, while the third returns its mean). We found it necessary to add some level of variance, especially in the early stages of optimization (we set $\tau_{gumbel}=2$). This ensures that learning can “explore” and obtain information regarding features and/or masks that have low probability under the current $\pi$.
> >
> > 2. Replacing the top-k operator with a soft top-k operator:
> > Given $\theta$, the soft top-k operator works by shifting some weight from the top-k features to the other, lower-valued features, which is necessary for gradients to be informative of all features. The implication is that if $\mu$ encodes a “good” mask, then its value as considered by the objective is potentially lower than its true test-time value. We have observed that only mild smoothing is necessary to facilitate effective optimization (we set $\tau_{top-k}=0.2$).
> >
> > 3. Replacing 0-1 predictions with probabilistic predictions:
> > This is the standard form of smoothing used in supervised learning (e.g, as when learning using log-loss). Here, we have observed that optimization succeeds with very little smoothing (we set $\tau_f=0.01$).

---

### Official Review · Reviewer_gK2B · 2023-10-24

**Soundness:** 3 good
**Presentation:** 2 fair
**Contribution:** 3 good
**Rating:** 6
**Confidence:** 3

**Summary:**

This work is devoted to improving welfare of users (buyers) in marketplaces of goods (like Yelp, AirBnb, etc). The authors propose to translate this problem into the problem of adjusting representations of goods (items) in the way it improves welfare through reducing congestion per each item. The representations are binary masks over item features (a single mask per market; so, no discrimination of users or items). The authors propose learn user preferences through a dataset obtained in the past. Extensive experimentation is done to justify applicability of proposed approach.

**Strengths:**

-	Original and novel work, interesting setup

-	Huge experimentation (most part is deferred to Appendix)

-	Practical applicability of the solution

**Weaknesses:**

-	Argumentation of the setup

-	Presentation

-	Details on ML setting


(see Questions field)

**Questions:**

1.	Argumentation of the setup:

a.	In Abstract “The power of a platform is limited to controlling representations— the subset of information about items presented by default to users”. This statement is very strong and seems not true. For instance, platforms definitely have other means to control information: besides representations (the amount of info provided per item) there are different ways to control user attention between different items like ranking of items, recommendation of items, etc. So, I strongly suggest rewriting this sentence.

b.	In Intro, the end of 2nd paragraph and 3rd paragraph: I do not understand why the described here issue cannot be resolved by some auction (or other mechanism design). The way these paragraphs are written, it sounds like the authors are not aware of vast practical application of auction in web services:

•	see, e.g., ad auctions, where they are built to reduce congestion by maximizing welfare (e.g., second price auctions, position auctions) through exploiting imbalance between demand and supply; and in this case, prices (bids) are also set in decentralized way – so, this argument, does not imply strong conclusion that representation is the only one way.

c.	In Intro, Page 2, 3rd paragraph: “under perceived values remain both valuable and diverse.” Why? + Example after this does not help and is unclear. Why does the problem cannot be resolved by auctions (so, the platform adjust price despite its decentralized price input) or by ranking (playing user attention)?

d.	I strongly recommend reviewing and rewrite argumentation of viability of the proposed setup: the setup itself sounds, but it should not be positioned as the only way for resolve the marketplace / platform issues…

---------

2.	Presentation: I believe Intro can have more details (preserving the same space). For instance,

a.	While reading the whole Intro, for me, it was still unclear what is meant by “representation”: whether representation is dependent on item (in Setup I’ve found it is not), whether it is about smth like ranking or so (in Setup I’ve found it is not)

b.	Welfare: is it just about users? Or users + platform? Or welfare of users + sellers? Only, in Setup, I’ve found that it is about users only. BUT in Intro, it is written “reduced social welfare—to the detriment of users, suppliers, and the platform itself” which is misleading…

c.	Section 4: it would be nice to see some practical examples where these conditions are working.

---------

3.	ML

a.	In Intro, Page 2, 2nd paragraph: “Given this, we propose to use machine learning to solve the necessary design problem of choosing beneficial item representations.” Is it true that ML usage is a separate contribution in addition to the proposed setup? (Or ML is a part of the setup?) It would be nice to have clear list of the contributions.

b.	I believe it is better to improve Sec.2 by removing discussion of ML from discussion of the setup (and discuss ML in a separate section or introduce it directly in Sec.3). Right now, it creates a mix of not fully discussed ML injection (2nd and 3rd paragraphs Page 4: there is nothing about learning objective, what to learn \mu? \beta? etc) and Problem setup. I believe optimization problem (Welfare without Expectation) should be introduced before stepping into discussion of ML.

c.	Eq (6) and (7): Is it OK to train both f on S and use the same S for W_S calculation? Should we split S? Despite theoretical analysis of non-ML setup in Sec.4, I struggle from not having any guarantees on viability of proposed ML approach (are we using standard ML setting with well-known theory and practice?).  It would be nice to have such a discussion somewhere in the text.

---------

4.	Experiments:

a.	How are V_{het} and V_{hom} formally defined?

b.	It would be nice to have formal direct link/ref to Appendix where details of experimentation are given (both for Sec.5.1 and for Sec.5.2). It is important for reproducibility.

c.	In Sec.5.2. “we optimize Eq.(8) using Adam with…”. What is “Adam”? It is better to give better ref / citation, or more convenient naming.

---------

5.	Minor:

a.	Footnote 3: it would be interesting to know which problem we face (or which assumptions are broken) when sellers adapt quickly (e.g., dynamic pricing is very common setup in ad auctions)

b.	Eq.(7): formally, it is incorrect to write M \in S since S consists of pairs (M, y). So, either need to replace by (M,y)\in S or by a sum over l = 1,…,L

c.	In Sec.6: “"..as well as the study of more elaborate user.."”:  “of” --> “to” ?

---

> ### Author Response · Authors · 2023-11-19
> **Response (part 1/2)**
>
> Thank you for your detailed response and helpful comments. We are hopeful that our response below helps in addressing your questions and comments.
>
> **1. Argumentation of the setup:**
>
> **a. In the abstract: “statement is very strong… for instance, platforms definitely have other means to control information; … I suggest rewriting this sentence”**
> Thank you for this suggestion. We agree and already make this point later in the introduction (4th paragraph). We will make sure this message is also conveyed in the abstract.
>
> **b-d: Comments on our setup vs. auctions:**
> Thank you for this comment, please let us clarify: We agree that the general problem of optimizing welfare admits many possible approaches (auctions being one of them). True, if we were to design a *new* platform, or allowed to drastically change a current platform’s market mechanism, then an alternative would be to implement an auction mechanism - or even have the platform itself set prices.
>
> However, our work asks: how can a platform that *already* operates a market with seller-mediated prices improve welfare? We argue that *given this setup* - which is very common - it is inevitable that there is a choice to make about item representation, and that this provides an unexplored degree of freedom through which a platform can improve welfare. Thus, the “inevitability” we discuss concerns the *informational* aspect - not the overall market design choice. We will clarify this.
>
> Our choice to focus on platforms that operate “conventional” price-based (and seller-mediated price based) markets stems from their popularity: consider Airbnb, Zillow, Etsy, Upwork, AliExpress, Wayfair, and many others - all of which are consumer-facing and deal with complex products or services that must match with complex consumer preferences. And while auctions may be common for other domains (e.g., ads), they are far less so for consumer-facing platforms (Ebay is perhaps the only exception, and it too enables direct price setting be sellers).
>
>
> **2. Presentation:**
>
> **a. In the intro, it is unclear what is meant by “representation”.**
> Thank you for your comment. We will make clear the precise meaning of representation in the introduction (as you note, this is formally defined in Sec. 2).
>
> **b. Welfare: is it just about users? Or users + platform? Or welfare of users + sellers?**
>
> We work with the standard definition of social welfare, is which is the sum of utility to consumers *plus* the sum of revenue to sellers. A consumer’s utility is the value of the allocated item minus its price. The revenue to a seller is the price of the item sold. Because payments are transferred from buyers to sellers, these terms cancel out in the total sum:
>
> $W = \sum_i \sum_j a_{ij} (v_{ij} - p_j) + \sum_j \sum_i a_{ij} p_j = \sum_i \sum_j a_{ij} v_{ij},$
>
> which is precisely Eq. (3). We will clarify this in the final revision.
>
> Hence, welfare does consider utility to both buyers and sellers, while being agnostic to the way in which utility is distributed across different participants. We adopt welfare to users as the goal of the platform and this reflects a platform that is motivated by long-term engagement and satisfaction of users (and will seek to gain a share of transaction volume through transaction fees and/or other indirect forms of monetization).
>
> **c. Section 4: it would be nice to see some practical examples where these conditions are working.**
> The strongest condition (in Proposition 1) applies to markets in which the absolute differences in valuations (across items and users) is small relative to the magnitude of values. For example, consider a set of apartments in a certain location and of similar size, whose prices may differ due to different characteristics (such as layout or precise location), but where these differences are minor relative to apartments’ generally large `base’ price.
>
> The other conditions (1-5) aim to capture more subtle aspects of how the different characteristics behave. In particular, when does the representation of a limited number of features maintain efficient choices by users? In the apartments example, if all of the apartments in the market have similar, average-sized windows (condition 1), or if all users attribute sufficiently low value to the feature “has curtains” (condition 2), then an optimal allocation can be supported even while dropping these features in the item representation.

---

> > ### Author Response · Authors · 2023-11-19
> > **Response (part 2/2)**
> >
> > **3. ML**
> >
> > **a+b. “Is it true that ML usage is a separate contribution in addition to the proposed setup?”**
> > Yes, we believe that our setup and our approach (which is ML-based) are distinct contributions - the former conceptual, and the latter practical. In principle, we agree that one could attempt to solve Eq. (3) by employing some discrete optimization algorithm directly over masks, and without training or using a predictive model of user choice. However, this would not make use of readily-available past market data, and is a challenging bi-level combinatorial optimization problem even for a single market (Eq. (3)).
> >
> > **c. “Is it OK to train both f on S and use the same S for W_S calculation?”**
> > Yes - we do not see a reason why this approach is not sound.
> >
> >
> > **4. Experiments**
> >
> > **a. “How are V_{het} and V_{hom} formally defined?”**
> > Please see Appendix D for a detailed description.
> >
> > **b. Direct reference to Appendix with details of experimentation**
> > Complete experimental details are given in Appendix D (for Sec. 5.1) and Appendix E (for Sec. 5.2). We will note this in the main text.
> >
> > **c. What is “Adam”?**
> > Adam is a highly-popular gradient-based optimization method due to Kingma and Ba (2014). Thank you for this suggestion, we will add an appropriate citation.
> >
> >
> > **5. Minor**
> >
> > **a. “It would be interesting to know which problem we face (or which assumptions are broken) when sellers adapt quickly”**
> > The primary challenge is that adaptive prices introduce a feedback loop: we would need to optimize representations that account for prices that are in turn affected by those very same representations. For a lengthy discussion, please see Appendix A.
> >
> > **b. “formally, it is incorrect to write M \in S.”**
> > This is true, we will fix this in the next revision.

---

### Official Review · Reviewer_Gepo · 2023-11-01

**Soundness:** 3 good
**Presentation:** 3 good
**Contribution:** 3 good
**Rating:** 8
**Confidence:** 2

**Summary:**

This paper initiates the study of decongestion by representation in the setting that f a platform is limited to controlling representations— the subset of information about items presented by default to users. A differentiable learning framework is developed to learn item representations in order to reduce congestion and improve social welfare. It is shown that partial information is a necessary aspect of modern online markets, and that systems have both the opportunity and responsibility in choosing representations that serve their users well.  Sufficient conditions for when decongestion promotes welfare are developed. Extensive experiments on both synthetic and real data demonstrate the utility of the proposed approach.

**Strengths:**

This paper formulates an interesting problem of decongestion by representation, which has great practical value.

The proposed differentiable learning framework looks sound and yield insightful results.

The theoretical analysis looks sound and it is supplemented by extensive experiments.

**Weaknesses:**

I am not an expert of this paper.  I do no identify any major weaknesses of this paper.

**Questions:**

No questions.

**Details Of Ethics Concerns:**

No.

---

> ### Author Response · Authors · 2023-11-19
>
> Thank you for your encouraging review! We were delighted that you found our paper to be interesting and of practical value. Should any questions arise, we would be happy to respond.

---

### Official Review · Reviewer_7NAi · 2023-11-02

**Soundness:** 3 good
**Presentation:** 3 good
**Contribution:** 3 good
**Rating:** 8
**Confidence:** 3

**Summary:**

This paper addresses the issue of market congestion, where consumers often compete inefficiently for the same subset of goods or services. To alleviate the issue, the authors propose "decongestion by representation," where a platform learns to display item information in ways that reduce congestion and improve social welfare. The key of the approach is a "differentiable proxy of welfare", which enables an end-to-end training process based on consumer choice data. Extensive experiments on both synthetic and real data show the effectiveness of the approach.

**Strengths:**

- The study is highly relevant to modern e-commerce platforms, potentially leading to better consumer experiences and more efficient markets.
- The idea of resolving market congestion through selective information representation is novel and addresses real-world concerns in online marketplaces.
- The differentiable proxy for welfare approach is sound and technically rigorous, which provides a strong analytical foundation of the proposed solution.

**Weaknesses:**

- The discussion in the ethics statement does not really relieve my concern that the manipulation of representations would open up the Pandora's box for online recommender platforms. The same approach can be applied to optimize user welfare but can also be exploited for promoting the revenue which might hurt the user satisfaction.

**Questions:**

- The core of the optimization technique is to replace the welfare function with a lower bound proxy. I'm curious how tight Eq (5) is? It would be nice to add some discussions in this regard.

- The experiment result in Figure 4 seems to suggest that a larger mask size $k$ leads to a higher welfare gain when $d$ is large. I'm not sure why it is possible: since a larger $k$ induces a loss in the perceived value so there should be a trade-off between perceived value and congestion level. I'm expecting an inverted-U curve and the result seems counterintuitive to me. Could you explain what I'm missing here?

---

> ### Author Response · Authors · 2023-11-19
>
> Thank you for your positive review! We were happy to learn that you found our work to be relevant and novel.
>
>
> **Is Eq. 5 tight?**
> Generally, our proxy welfare objective need not be tight, which is expected given that it does not include unobserved valuations (whereas true welfare does). One case in which our proxy is tight is when preferences are fully heterogeneous and prices are seller-optimal CE. These conditions ensure that term (II) in Eq. (4) is zero (i.e., there is no congestion) and term (I) aligns with welfare. This is analogous to proxy losses for classification: hinge loss and log-loss are useful as proxies to 0-1 loss and exact in simple special cases (e.g., separability).
>
> **”Larger mask size k leads to a higher welfare gain when d is large. I'm not sure why it is possible.”**
> Please note that Fig. 4 displays welfare *gain* relative to the random baseline, and the differences between d=100 and d=12 are relative to this baseline. From the overlay numbers (and also Fig. 12 in the Appendix) you can also see, for all k<d, that the *absolute* welfare gain increases with k for all methods and irrespective of whether d= 12 or d=100.
>
>
> **“The same approach can be applied to optimize user welfare but can also be exploited for promoting the revenue which might hurt the user satisfaction.”**
> We haven’t experimented with this, but agree that it’s possible. Still, this is no different from the tradeoffs that platforms always face. How to balance user welfare (“social welfare”) with platform profit is a question of balancing long-term engagement of users with short-term platform gain and mediated by questions related to platform competition, platform growth, and regulation.
>
> We also note that our approach could likely allow a platform to optimize welfare *alongside* revenue. In classic market theory, there are typically multiple choices of equilibrium (CE) prices - all of which enable maximal welfare, but can differ in revenue (Shapley and Shubik, 1971). If the platform controls prices, then one choice is to select seller-optimal CE prices. When the platform does not control prices, a possible extension of our framework could be to learn representations that maximize revenue *subject to welfare constraints*, or optimize some combination of welfare and revenue.

---

> > ### Comment · Reviewer_7NAi · 2023-11-23
> > **Response to the authors**
> >
> > Thanks for the response. For my second question, maybe I was not expressing my concern clearly: what I really want to ask is, could you provide an explanation why the absolute welfare gain shows different trends: it almost always increases with k when d=100 and it increases and then decrease with k when d=12?

---

### Meta-Review · Area_Chair_7F2F · 2023-12-02

**Metareview:**

This paper studies the problem of decongestion by representation, designing the information representations for items in online market places to improve social welfare. Through a differential proxy for the objective, the authors propose to learn the design of representations from past customer data. The proposed approach is examined through extensive experiments both on simulated data and real data.

Overall all reviewers agree that the paper is addressing a practically relevant question in online market places. The proposed approach is sound and the evaluations are generally convincing. There have been various comments and suggestions regarding the framing and presentations of the paper, however, they all seem addressable in the revisions, as demonstrated in the author responses.

**Justification For Why Not Higher Score:**

The main novelty of this paper arises from how the research question is framed. The overall approach seems relatively standard, except possibly for the differential proxy for utility.

I am currently recommending this for a poster presentation. However, depending on the level of interest in the topic within the ICLR community, it might be promoted to a spotlight.

**Justification For Why Not Lower Score:**

The paper is overall solid, with an interesting research question and well-executed evaluations.  Given the overall positive reviews, it should appear in ICLR proceedings.

---

### Decision · Program_Chairs · 2024-01-16

Accept (poster)